# Comparison of satellite based evapotranspiration estimates over the Tibetan Plateau

Jian Peng[1*], Alexander Loew[1,5], Xuelong Chen[2], Yaoming Ma[3,4], Zhongbo Su[2]

[1] Max Planck Institute for Meteorology, 20146 Hamburg, Germany;

[2] Faculty of Geo-Information Science and Earth Observation, University of Twente, Enschede 7500 AE, the Netherlands;

[3] Key Laboratory of Tibetan Environment Changes and Land Surface Processes, Institute of Tibetan Plateau Research, Chinese Academy of Sciences, Beijing 100101, China;

[4] CAS Center for Excellence in Tibetan Plateau Earth Sciences, Chinese Academy of Sciences, Beijing 100101, China;

[5] Department of Geography, Ludwig-Maximilians Universität München (LMU), 80333 Munich, Germany

*Corresponding author: Tel: +49-(0)-89-2180-6515; Fax: +49-(0)-40-41173-350.

E-mail addresses: jian.peng@mpimet.mpg.de

## Abstract

The Tibetan Plateau (TP) plays a major role in regional and global climate. The knowledge of latent heat flux can help to better describe the complex mechanisms and interactions between land and atmosphere. Despite its importance, accurate estimation of Evapotranspiration (ET) over the TP remains challenging. Satellite observations allow for ET estimation at high temporal and spatial scales. The purpose of this paper is to provide a detailed cross comparison of existing ET products over the TP. Six available ET products based on different approaches are included for comparison. Results show that all products capture the seasonal variability well with minimum ET in the winter and maximum ET in the summer. Regarding the spatial pattern, the High Resolution Land Surface Parameters from Space (HOLAPS) ET demonstrator dataset is very similar to the LandFlux-EVAL dataset (a benchmark ET product from the Global Energy and Water Cycle Experiment), with decreasing ET from the southeast to northwest over the TP. Further comparison against the LandFlux-EVAL over different sub-regions that are decided by different intervals of normalized difference vegetation index (NDVI), precipitation and elevation reveals that HOLAPS agrees best with LandFlux-EVAL having the highest correlation coefficient (R) and lowest Root Mean Square Difference (RMSD). These results indicate the potential for the application of the HOLAPS demonstrator dataset in understanding the land-atmosphere-biosphere interactions over the TP. In order to provide more accurate ET over the TP, model calibration, high accuracy forcing dataset, appropriate in situ measurements as well as other hydrological data such as runoff measurements are still needed.

**Keywords**: HOLAPS; Tibetan Plateau; Evapotranspiration; Latent heat flux; Water fluxes; Land-atmosphere interactions

# 1. Introduction

Evapotranspiration (ET) is an essential nexus of energy and water cycles through the mass and energy interactions between land and atmosphere (Jung et al., 2010; Peng et al., 2013a). The estimation of spatially distributed ET has been advanced by the progress of satellite remote sensing technology. However, remote sensing techniques do not allow to directly inverting ET from space (Peng et al., 2013b; Zhang et al., 2016a). Different methods have been therefore developed to estimate ET with the use of physical variables that are sensed by satellite and are related to the evaporation process (Kalma et al., 2008; Wang and Dickinson, 2012). In recent years, a number of global ET products have been generated with the availability of long-term global satellite products and progress in computer science (Zhang et al., 2010; Vinukollu et al., 2011j; Miralles et al., 2011; Fisher et al., 2008). Some of these global products can even provide ET with spatial resolution less than 10 km and temporal resolution less than 3 hour (Mu et al., 2007; Miralles et al., 2016; Loew et al., 2015). HOLAPS (High resOlution Land Atmosphere surface Parameters from Space) demonstrator dataset is one of them. HOLAPS is actually a framework that can provide surface energy and water fluxes at sub-hourly timescales and spatial resolutions at the kilometer scale. It is also worth noting that very high spatial resolution (on the order of 10 m) ET product at regional scale can be provided by ALEXI/DisALEXI based on thermal observations from polar and geostationary orbiting satellites (Anderson et al., 2011; Anderson et al., 2007). Although these global ET products have been applied to many applications such as multi-decadal trend analysis (Zhang et al., 2016b; Zhang et al., 2015; Miralles et al., 2014; Jiménez et al., 2011), large discrepancies remain exist in these products. Within the Global Energy and Water Cycle Experiment (GEWEX) LandFlux initiative, Mueller et al. (2011) conducted a comparison of existing global LE products from either land surface models, re-analysis, or satellite estimates, and found that the global mean LE over land was $45\pm5$ W/m$^2$, with a spread of 20 W/m$^2$. In addition, a synthesis dataset has also been generated within the GEWEX LandFlux-EVAL initiative, which provides LE at monthly timescale and a spatial resolution of 1 degree (Mueller et al., 2013). Recently, several studies have evaluated commonly used ET retrieval algorithms, including Penman-Monteith (PM) algorithm, the Priestley-Taylor (PT) model and the Surface Energy Balance System (SEBS) (Su, 2002), which are driven by the same forcing dataset at both FLUXNET tower and global scales (Vinukollu et al., 2011j; Miralles et al., 2016; Michel et al., 2016; McCabe et al., 2016; Ershadi et al., 2014). To develop a more accurate global LE product, improvements of the parameterization and sensitivity analysis of the model to forcing dataset are still needed (Michel et al., 2016; McCabe et al., 2016). Note that the energy equivalent for ET is referred as latent heat flux (LE), which is used interchangeable with ET in this paper.

Nevertheless, theses global ET products have great potentials for global and regional hydrological applications. In this study, the performances of the widely used global ET products will be investigated over the Tibetan Plateau (TP), as the ET over

the TP is of great importance and research interest. The TP has strong impacts on weather and climate at the regional to global scale and controls climatic and environmental changes in Asia and elsewhere in the Northern Hemisphere (Ma et al., 2008). The knowledge of ET is essential for the study of land-atmosphere interactions, and assessment of the impacts of and feedbacks to the global change (Shi and Liang, 2014). In order to characterize the distribution of ET over the TP, different methods using micrometeorological measurements (Yang et al., 2003; Lee et al., 2012; Chen et al., 2013b; Zhang et al., 2007), remote sensing products (Ma et al., 2014; Ma et al., 2006; Chen et al., 2013a) and the combined use of both data sources (Ma et al., 2003; Ma et al., 2011; You et al., 2014) have been investigated over the last decades. In addition, land surface models have also been applied to simulate ET over the TP (Gerken et al., 2012; Yang et al., 2009). However, accurate estimation of ET over TP is still a challenge due to the limitations of the above approaches. Specifically, the observation-based methods are not adequate for determination of regional ET due to the limited spatial representativeness of meteorological stations, while the remote sensing products are only available under clear sky conditions. The models results are limited by the accuracy of input parameters and the uncertainties of model parameterization over complicated topography and highly heterogeneous areas of the TP (Shi and Liang, 2013d). The existing global ET products especially those with high spatial and temporal resolutions such as HOLAPS provide a potentially applicable ET dataset over the TP. Although the global ET products have been validated against FLUXNET measurements, the reliability of spatial and temporal patterns of them over the TP is still unknown. A comprehensive analysis of the characteristics of the LE over the TP based on the state-of-the-art global ET products has not yet been conducted. Therefore, the main objective of this study is to provide a detailed cross comparison of the different existing ET products over the TP. Through this study, the following research questions will be addressed: (1) Do existing global ET products show consistent spatial and temporal patterns over the TP? (2) Are there systematic deviations between the different data products which can be explained by different climate or surface conditions? The study will focus mainly on a cross-comparison between the different existing dataset due to a lack of appropriate reference data in the region like will be discussed.

## 2. Study area

The Tibetan Plateau (TP), known as the third pole of the Earth (Qiu, 2008), covers approximately the latitude from 26º N to 40º N, and longitude from 75º E to 105º E, with an area of 2500000 km$^2$. It is the highest and largest plateau in the world, with very complex terrain and an average elevation higher than 4000 m above sea level (asl) (Figure 1) (Frauenfeld et al., 2005; Ma et al., 2008). Due to its unique and special geographical position and physical environment, the climate of TP is influenced by both Asian monsoon and westerlies (Yang et al., 2014), and it has profound thermal and dynamical impacts on atmospheric circulation over China, the whole East Asia and even the entire globe (Cui and Graf, 2009; You et al., 2014). Specifically, the TP

reaches the middle troposphere and influences the atmospheric circulation through
mechanical forcing (Yanai and Li, 1994). On the other hand, the thermal forcing of the
TP enhances the Asian summer monsoon and influences its variability (Duan and Wu,
2005; Lau et al., 2006). In addition, the melting water from snow and glaciers in TP is
the source of many rivers in South and East Asia such as Yangtze,
Ganges-Brahmaputra. Therefore, the TP is also known as 'the Asian water tower',
supporting approximately 25% of the world's population (Immerzeel et al., 2010; Xu
et al., 2008). Quantitative estimation of the water and energy cycles over the TP is of
great significance for the study of land-atmosphere-biosphere interactions, and
understanding its response to climate change. (Sellers et al., 1997; Yang et al., 2014).

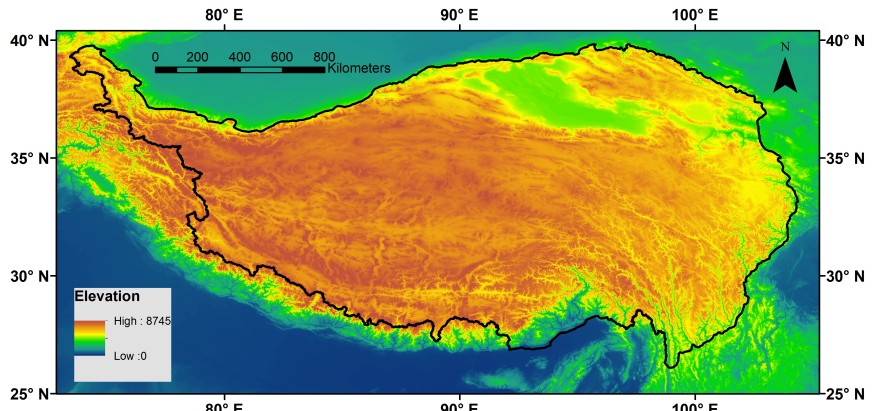

Figure 1: Map of the location and topography of the Tibetan Plateau.

## 3. Data and methods

### 3.1 Data

Different groups of algorithms have been developed to estimate ET from satellite data.
These comprise (1) surface energy balance models forced either by satellite remote
sensing or re-analysis data (Bastiaanssen et al., 1998; Su, 2002); (2) the methods based
on Penman-Monteith (PM) or Priestley and Taylor (PT) equations (Fisher et al., 2008;
Miralles et al., 2011; Mu et al., 2007; Zhang et al., 2015); (3) spatial variability
methods (Peng et al., 2013b; Peng and Loew, 2014; Roerink et al., 2000). Among them,
the PM algorithm, the PT model and the Surface Energy Balance System (SEBS) are
widely used, and have been explored by both GEWEX LandFlux-EVAL initiative and
the Water Cycle Multi-mission Observation Strategy EvapoTranspiration
(WACMOS-ET) project. Therefore, three LE datasets based these models and driven
by same forcing data are compared over the TP in this study. These datasets are
$SEBS_{SRB-PU}$, $PT_{SRB-PU}$ and $PM_{SRB-PU}$, which are respectively based on SEBS, PT, and
PM algorithms but driven by the same input radiation from Surface Radiation Budget
(SRB) (Stackhouse et al., 2011) and meteorological forcing datasets from Princeton
University (PU) (Vinukollu et al., 2011a). These three datasets used in this study were
obtained from the Princeton University Terrestrial Hydrology Research Group. In
addition, to investigate the impact of forcing data on the estimation of LE, another
recent released SEBS dataset (SEBS$_{Chen}$) is also included in this study (Chen et al.,
2014). Different from SEBS$_{SRB-PU}$, SEBS$_{Chen}$ is driven by the meteorological forcing
data obtained from the Institute of Tibetan Plateau Research, Chinese Academy of
Sciences (ITP, CAS), which was generated based on 740 weather stations operated by
the China Meteorological Administration. In addition, the recently developed
HOLAPS LE demonstrator dataset is also included for comparison. A brief description
of these products is presented below. For detailed algorithms and parameterizations of
these datasets, the readers are referred to the original articles: SEBS$_{SRB-PU}$, PT$_{SRB-PU}$
and PM$_{SRB-PU}$ (Vinukollu et al., 2011a), SEBS$_{Chen}$ (Chen et al., 2014), and HOLAPS
(Loew et al., 2015).

*SEBS:* SEBS is a one-source energy balance algorithm, which firstly calculates the
sensible heat flux (H) based on the Monin and Obukhov theory (Monin and Obukhov,
1954) with the requirement of surface temperature, air temperature gradient and the
parameterization of aerodynamic resistance. To constrain H within a lower and upper
boundary, two limiting conditions are considered. Under dry limit, the ET is equal to 0
and H is at its maximum, while the ET reaches to its potential rate and H is at its
minimum under wet limit. After the H is calculated, ET can be obtained through
closing the energy balance with the availability of net radiation and ground heat flux.
SEBS has already been widely validated with ground-based measurements over
different areas. Two SEBS datasets are included in the comparison. The SEBS$_{SRB-PU}$
was generated by Vinukollu et al. (2011a) and based on radiation from Surface
Radiation Budget (SRB) and meteorological forcing datasets from Princeton
University (PU) (Vinukollu et al., 2011a), while SEBS$_{Chen}$ estimated ET with
meteorological forcing data from the Institute of Tibetan Plateau Research, Chinese
Academy of Sciences (ITP, CAS). The monthly SEBS$_{Chen}$ ET has been found to agree
well with ground-based measurements over China (Chen et al., 2014). The comparison
of these two SEBS datasets can show the impact of forcing dataset on the estimation of
LE for the same type of model.

*PM$_{SRB-PU}$:* The PM$_{SRB-PU}$ is estimated based on a revised PM model (Mu et al.,
2007; Mu et al., 2011), which has been widely used to estimate global ET. Due to its
basis of Penman-Monteith equation, the PM model has high demand of inputs, with
high-level parameterization of the aerodynamic and surface resistances using
meteorological data and vegetation phenology. In contrast to the most PM based ET
models, two improvements have been implemented in PM$_{SRB-PU}$: (1) instead of a fixed
value, a biome-specific value for the mean potential stomatal conductance is applied;
(2) the aerodynamic resistance parameterization used by SEBS is applied here to
account for wind speed and boundary layer stability (Vinukollu et al., 2011a). The
PM$_{SRB-PU}$ is based on the same forcing data as SEBS$_{SRB-PU}$.

*PT$_{SRB-PU}$:* The PT-JPL model by Fisher et al. (2008) is used to estimate PT$_{SRB-PU}$.
Different from the PM model, the PT model does not require the parameterization of
the aerodynamic and surface resistances. Traditionally, the Priestley-Taylor (PT)
equation (Priestley and Taylor, 1972) is used to estimate potential ET, while the
PT-JPL model adjust it to estimate actual ET through considering ecophysiological
stress factors based on atmospheric moisture and vegetation indices. This implies that
the forcing data required for $PT_{SRB-PU}$ is quite comparable to that of $PM_{SRB-PU}$. The
$PT_{SRB-PU}$ relies on the same forcing datasets as $SEBS_{SRB-PU}$ and $PM_{SRB-PU}$, which
provides the possibility to investigate the performance of different ET models driven
by the same forcing data over the TP.
*HOLAPS:* The HOLAPS LE product was generated from HOLAPS framework,
which makes use of meteorological drivers coming exclusively from globally available
satellite and re-analysis datasets and is based on a state-of-the-art land surface scheme
(Loew et al., 2015). It is based on a radiation module, a planetary boundary layer
model, a soil module and a general module for the exchange of energy and moisture at
the surface layer. HOLAPS can ensure internal consistency of the different energy and
water fluxes and provide estimates at high temporal (< 1h) and spatial (~5 km)
resolutions. Good agreement with *in situ* measurements have also been found by Loew
et al. (2015) when compared against 48 FLUXNET stations worldwide. The details of
the HOLAPS framework and relevant evaluation results can be found in the reference
of Loew et al. (2015).
The validation of different LE datasets against in-situ measurements over the TP is
not possible for the current study period due to: a) the access to suitable *in situ*
measurements is not possible; b) spatial representativeness of the existing FLUXNET
towers for areas of only several square kilometers. Therefore, the above LE datasets
are cross-compared with LandFlux-EVAL benchmark product in the current analysis.
LandFlux-EVAL is a merged synthesis LE product based on a total of 14 datasets
including land surface model output, observations-based estimates, and atmospheric
reanalyses (Mueller et al., 2013). It provides the best guess estimate of LE for the first
time based on the existing global LE datasets, and also provides the uncertainty range
of the absolute LE values (interquartile range of the merged synthesis LE products).
Note that the merged LE dataset agreed well with precipitation minus runoff over large
river basins around the world (Mueller et al., 2011), and it has been used to evaluate
the LE simulations of the fifth phase of the Coupled model Inter-comparison project
(CMIP5) (Mueller and Seneviratne, 2014). To further demonstrate the validity of
LandFlux-EVAL benchmark product over the TP, we also compared it to precipitation,
which is one of the most important driving factors for LE. It should be noted here that
LandFlux-EVAL also includes satellite-based LE datasets that are estimated from PM
and PT algorithms. However, the $PM_{SRB-PU}$ and $PT_{SRB-PU}$ datasets used in the current
analysis are different from those datasets. They are based on revised PM and PT
approaches, which also account for the evaporation from canopy intercepted
precipitation (Vinukollu et al., 2011a). In addition, the forcing datasets used for
$PM_{SRB-PU}$ and $PT_{SRB-PU}$ are also different from that used for PM and PT datasets in
LandFlux-EVAL. For example, the radiation used for the $PM_{SRB-PU}$ is from SRB, while
PM dataset from LandFlux-EVAL uses radiation from International Satellite Cloud
Climatology Project (ISCCP). A summary of these datasets is given in Table 1. For
detailed information about each product, the reader is referred to the relevant
publications.
Table 1: Summary of the datasets used in our study.

| Dataset | ET scheme | Spatial resolution | Temporal resolution | Reference |
|---|---|---|---|---|
| $PM_{SRB-PU}$ | Penman-Monteith | 1º | daily | *(Vinukollu et al., 2011a)* |
| $PT_{SRB-PU}$ | Priestley-Taylor | 1º | daily | *(Vinukollu et al., 2011a)* |
| $SEBS_{SRB-PU}$ | SEBS | 1º | daily | *(Vinukollu et al., 2011a)* |
| $SEBS_{Chen}$ | SEBS | 0.1º | daily | *(Chen et al., 2014)* |
| HOLAPS | Priestley-Taylor | 5 km | half hourly | *(Loew et al., 2015)* |
| LandFlux-EVAL | Synthesis product | 1º | monthly | *(Mueller et al., 2013)* |


### *3.2 Methods*

*3.2.1 Data Preprocessing*
All of the datasets were firstly aggregated to monthly mean values over the common
time period 2001-2005, which corresponds to the temporal resolution of
LandFlux-EVAL benchmark product and the time period currently covered by the
HOLAPS demonstrator dataset (Loew et al., 2015; Mueller et al., 2013). To make an
unbiased comparison with LandFlux-EVAL dataset, HOLAPS and $SEBS_{Chen}$ were
further aggregated to the same spatial resolution as LandFlux-EVAL. In addition, the
current HOLAPS demonstrator dataset does not include the estimate of LE over
snow-covered areas. Therefore, the snow-covered areas of all the products were also
masked out based on the MODIS snow cover product.
*3.2.2 Spatial and temporal analysis*
The characteristics of all the datasets were investigated through spatial and temporal
analysis. The spatial distributions of the seasonal and annual average LE over the TP
were analyzed, including the identification of patterns such as low and high values, and
the investigation of seasonal changes. The four seasons are defined as autumn
(September–October–November), winter (December–January–February), spring
(March–April–May), and summer (June–July–August). The temporal analysis explored
the seasonal and annual variation of all the datasets from 2001 to 2005 over the whole
TP. In addition, the correlation analysis was conducted to evaluate the impacts of
climate (precipitation) and surface conditions (normalized difference vegetation index
and elevation) on the performance of ET estimation. The relationship between different
LE products and the LandFlux-EVAL benchmark product were quantified by using
correlation coefficient and root-mean-square deviation over the whole TP and different
sub-regions, which were decided by different intervals of normalized difference
vegetation index (NDVI, generated from MODIS), precipitation (Global Precipitation
Climatology Project, GPCP) and elevation (Global Multi-resolution Terrain Elevation
Data 2010, GMTED2010).

## 4 Results and discussion


### 4.1 Spatial and temporal variability of different LE products


The spatial distributions of annual mean LandFlux-EVAL and precipitation are shown
in Figure 2. It can be seen that the LE has similar patterns as observation-based
precipitation, both decreasing from southeast to northwest over the TP. The
comparison of all the pixels shows a very high correlation coefficient of 0.9 between
LE and precipitation. Besides precipitation, the radiation is another important driver for
LE. Compared to the published studies, the LandFlux-EVAL LE also corresponds well
with the merged net radiation and LE datasets, which were developed and validated
over the TP by Shi and Liang (2013d, 2013a) and Shi and Liang (2014). The spatial
distribution of annual mean net radiation and LE can be found in study of Shi and
Liang (2013a) and Shi and Liang (2014). Although the LandFlux-EVAL has not been
validated against in-situ measurements over the TP, the similar spatial patterns
between LE and both observation-based precipitation and validated radiation to some
extent demonstrate the validity of LandFlux-EVAL over the TP.

Figure 3 displays the spatial pattern of annual mean values for different LE datasets.
Although these LE products have been reported performing well against FLUXNET
measurements at point scale, they exhibit differently in terms of spatial pattern over the
TP. In general, the LandFlux-EVAL, HOLAPS and SEBS$_{Chen}$ have high LE in the
southeastern TP and low LE in the northwestern TP, which might be related to the
decrease of elevation from northwest to southeast as well as the monsoon climate in
the southeastern TP. The spatial variations of PT$_{SRB-PU}$ and PM$_{SRB-PU}$ are related to the
increase of latitude from south to north, while SEBS$_{SRB-PU}$ has high and low LE in
outer and central TP.

Figure 4 further shows the annual mean spatial patterns of 25th-percentile and
75th-percentile of the LandFlux-EVAL multi-datasets ensemble, which quantifies the
uncertainty range of the absolute LE values (interquartile range of the merged
synthesis LE products). It can be seen that HOLAPS and most parts of PT$_{SRB-PU}$ and
PM$_{SRB-PU}$ are within the interquartile range, while outer part of SEBS$_{SRB-PU}$ and
southern part of SEBS$_{Chen}$ are out of the interquartile range. To make an unbiased
comparison between LandFlux-EVAL and other LE datasets, all the datasets were
resampled to the same spatial resolution as LandFlux-EVAL and masked out the
snow-covered areas. Figure 5 shows the differences of spatial patterns between
LandFlux-EVAL and other LE datasets. Overall, the HOLAPS dataset is found to have
good agreement with the benchmark product (LandFlux-EVAL) for most parts of TP.
The $PT_{SRB-PU}$ and $PM_{SRB-PU}$ are found to have positive biases over western TP, and
$SEBS_{SRB-PU}$ has bias over outer TP, and $SEBS_{Chen}$ has bias over southern TP.

328       Besides the analysis of spatial distribution of annual mean, the seasonal means of
each LE dataset are also show in Figure 6. It can be seen that all the LE datasets show
clear seasonal cycles with highest values in summer and lowest values in winter, which
might be related to both westerlies and Asian monsoon. Due to the influence of Asian
summer monsoon, the highest LE in LandFlux-EVAL is in southeastern TP and the LE
decreases to northwest. The lowest LE appears in northern TP where dry westerlies
dominate. Similar patterns are also found in HOLAPS, $PT_{SRB-PU}$, $PM_{SRB-PU}$ and
$SEBS_{Chen}$. The LE is lower in spring than that in summer in the eastern TP, which
relates to the onset of the Asian summer monsoon. All the datasets present very low
values during winter due to the cold and dry climate. The seasonal patterns of
LandFlux-EVAL are also consistent with the study by You et al. (2014), where the LE
was also found to increase from northwest to southeast in all seasons over the TP.
Overall, the HOLAPS is most similar to LandFlux-EVAL compared to other datasets
in terms of spatial distribution and spatial mean values over all seasons.


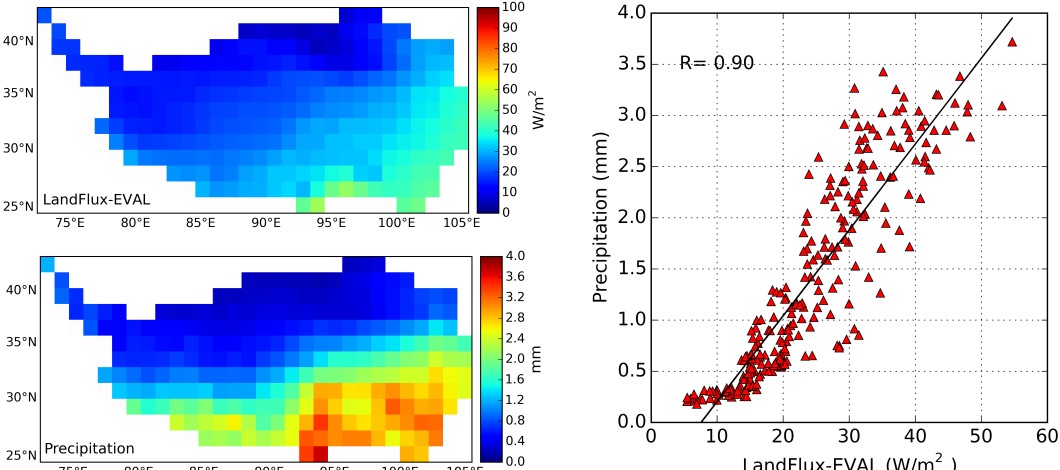


Figure 2. Spatial distribution of annual mean LandFlux-EVAL LE and GPCP
precipitation over the TP (left panel). The scatter plots of the comparison between LE
and precipitation for all the pixels (right panel).


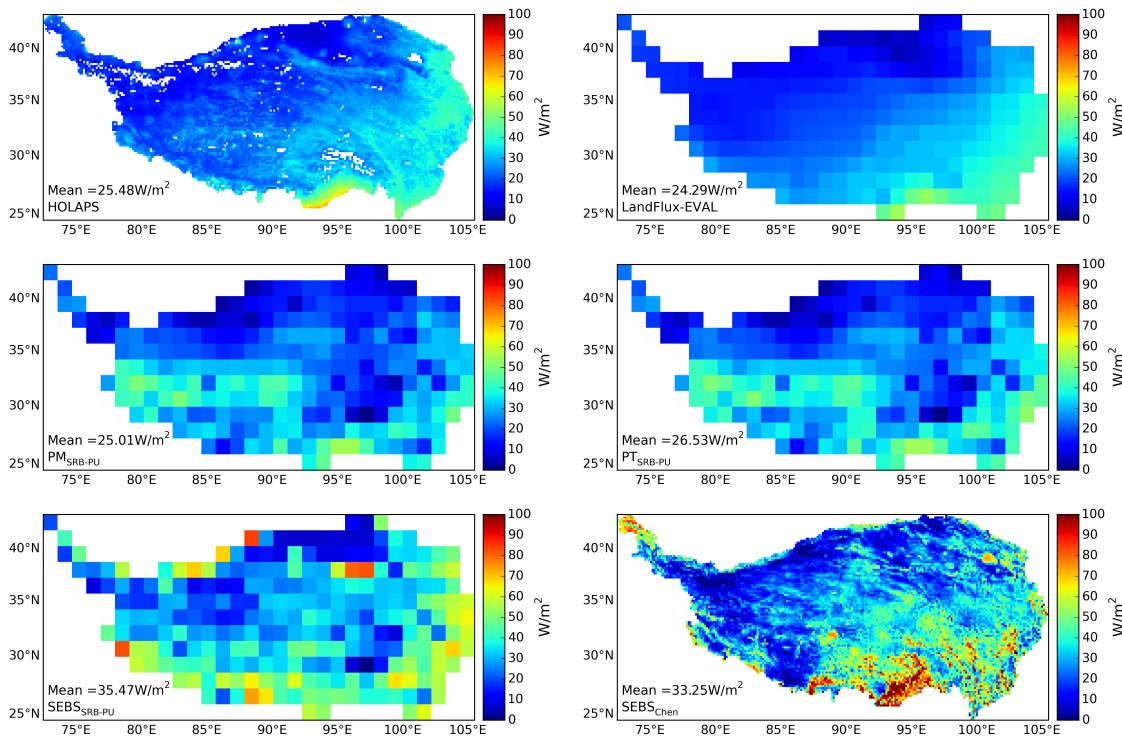


Figure 3: Spatial distribution of annual mean LE for each dataset over the TP.


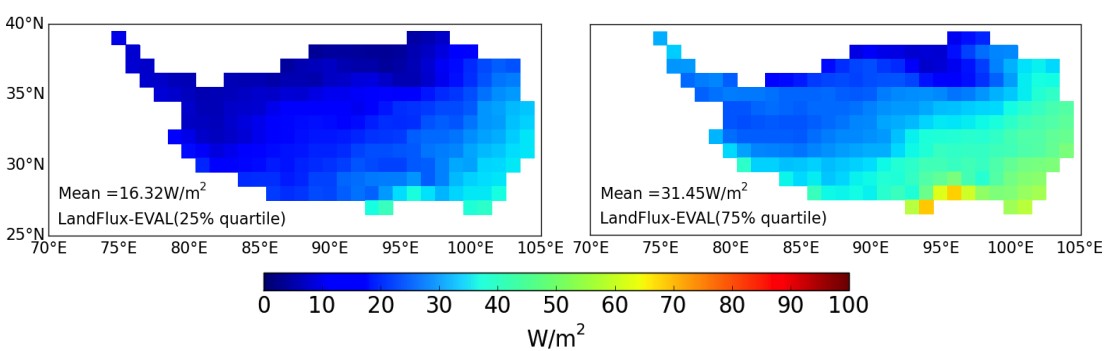


Figure 4: The annual mean spatial patterns of 25th-percentile and 75th-percentile of the
LandFlux-EVAL multi-datasets ensemble.


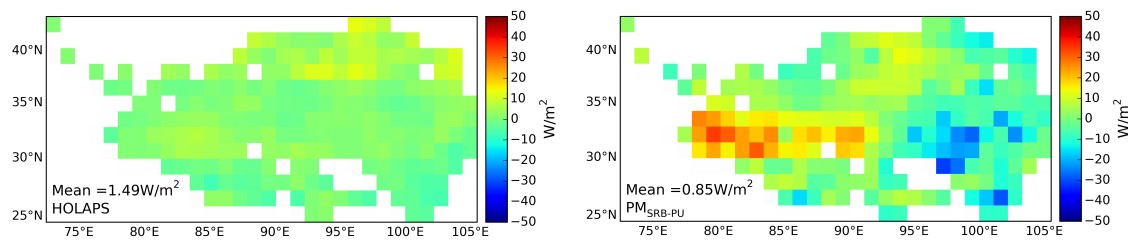

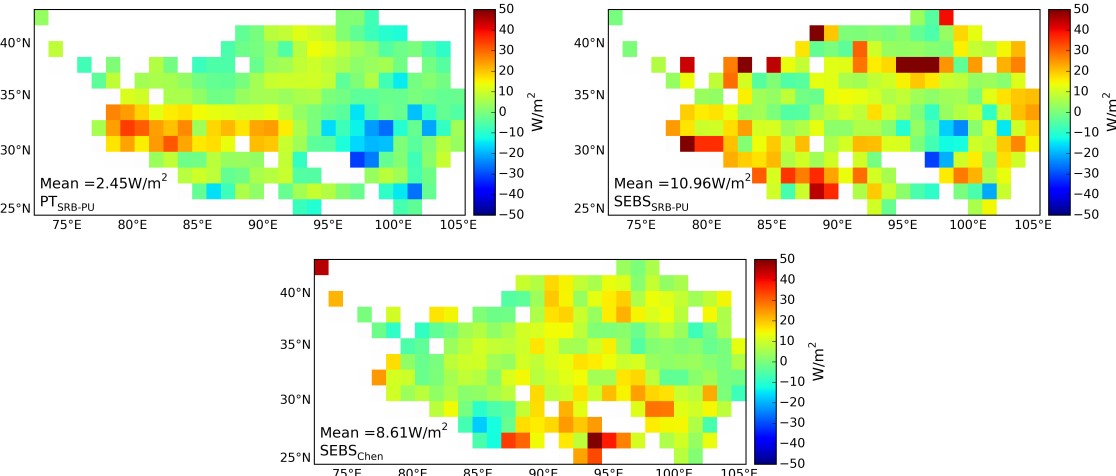

Figure 5: Differences of spatial distribution of annual mean LE between LandFlux-EVAL and other datasets over the TP.

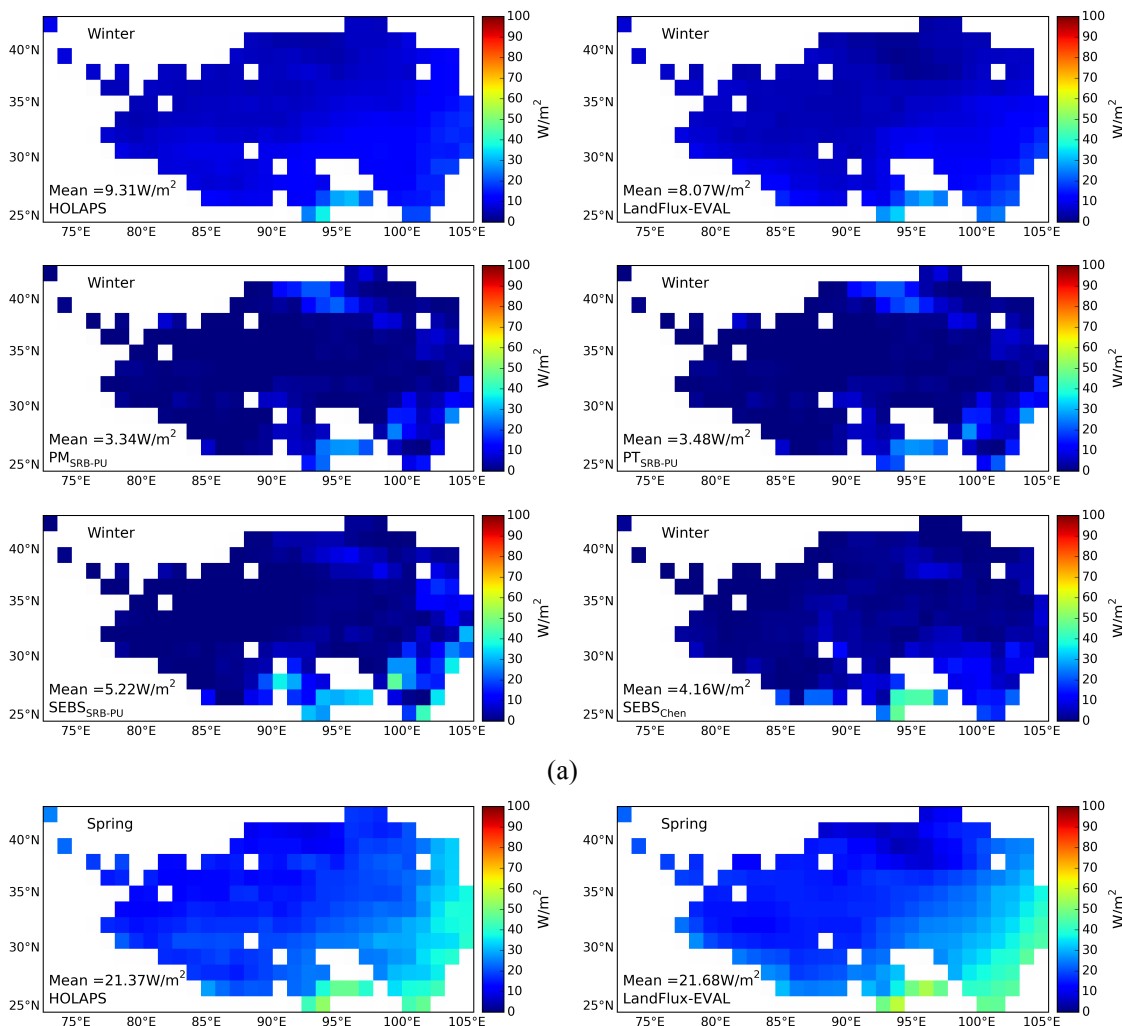

(a)

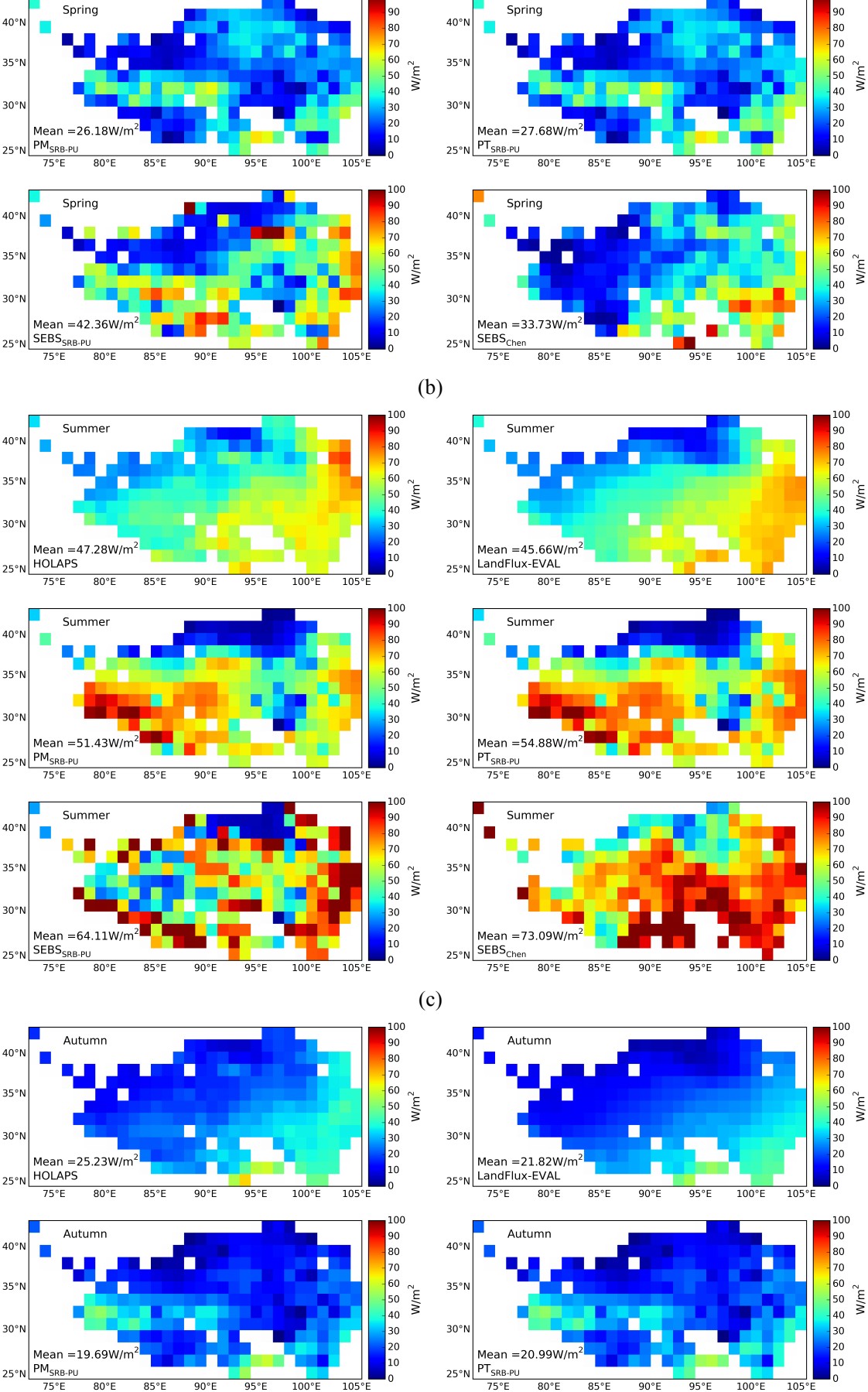

(b)

(c)

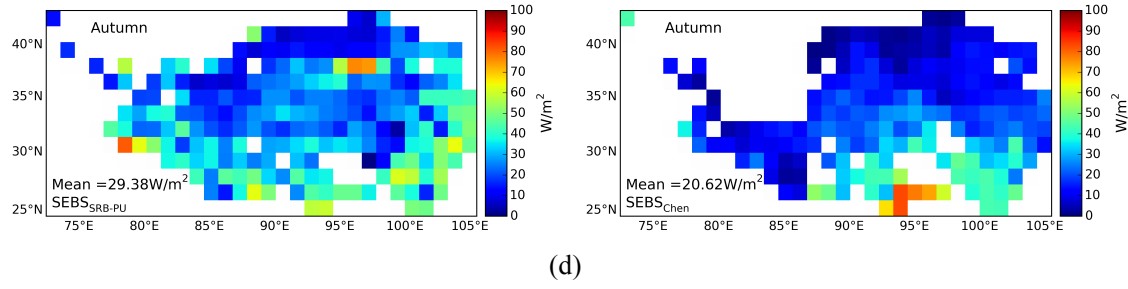

(d)

Figure 6: Spatial distribution of seasonal mean LE for each dataset over the TP. (a) Winter, (b) Spring, (c) Summer, (d) Autumn.

In addition to the spatial comparisons of annual and seasonal mean values, the time evolution of all datasets is also explored. Figure 7 presents the time series of the area mean LE for different LE datasets, and the inter-quartile range between 25th-percentile and 75th-percentile of the LandFlux-EVAL ensemble. According to Figure 7, all products capture well the seasonal variability with minimum LE in the winter and maximum LE in the summer. However, the mean values of different LE products differ substantially. There is a spread of about 35 $W/m^2$ at the annual cycle peak. Compared with the other products, the HOLAPS seems to be closer to the LandFlux-EVAL benchmarking product. The SEBS$_{SRB-PU}$ and SEBS$_{Chen}$ seem to be more distinctive with LE from most months outside the inter-quartile of LandFlux-EVAL ensemble. However, when compared to the climatology calculated from flux tower measurements around the TP the SEBS estimates seem to be close to the flux tower measurements (Chen, 2011). The differences between LandFluxEval and SEBS might be caused by the scale mismatch between gournd measurement at point scale and the satlellite estimate at pixel scale. The mismatch includes the surfacde heteogenity (such as topography, land cover types) and atmospheric conditions (such as cloud coverage, altitude variations) (You et al., 2014; Hakuba et al., 2013). Compared to SEBS (Chen, 2011), the LandFlux-EVAL has relative low spatial resolution of 1º, which might be strongly influenced by scale mismatch effects over comlex surface and atmospheric conditions in TP. Taking advantage of high temporal resolution of HOLAPS, the temporal variability of the area averaged LE for 5-day HOLAPS over the 2001-2005 is shown in Figure 8, where more temporal variations are found compared to monthly temporal variability. Besides, the temporal variation of the averaged LE over the TP has also been compared with precipitation and NDVI, which might regulate the LE. Table 2 shows the statistics of the comparisons. A strong correlation of higher than 0.7 has been found between all LE datasets and NDVI, implying the importance of vegetation on regulating LE over the TP. The highest R value was found between HOLAPS and NDVI. As expected, the LE has strong correlation to precipitation with R value higher than 0.87 for all LE datasets, which is because precipitation is one of the most important drivers for LE. In the next section, the performance of each product will be further discussed based on the comparison results against the LandFlux-EVAL benchmark product.

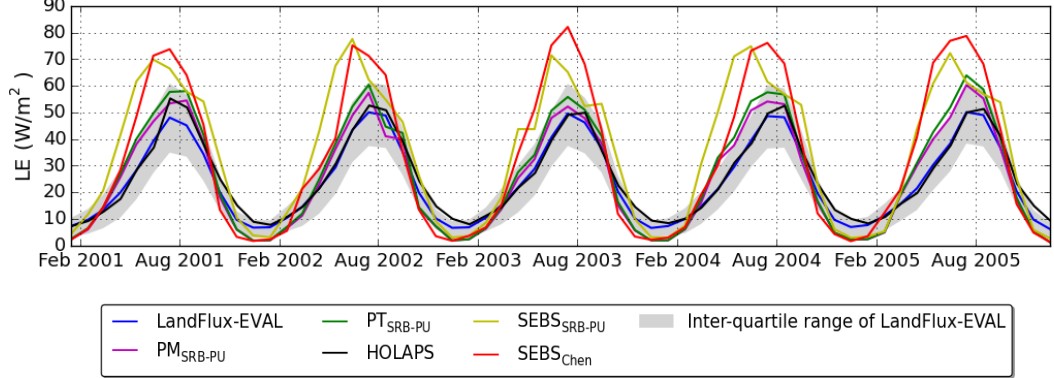


Figure 7: Temporal variability of the area averaged LE for each dataset over the TP. The grey
shadow displays the inter-quartile range between 25th-percentile and 75th-percentile of the
LandFlux-EVAL multi-datasets ensemble.


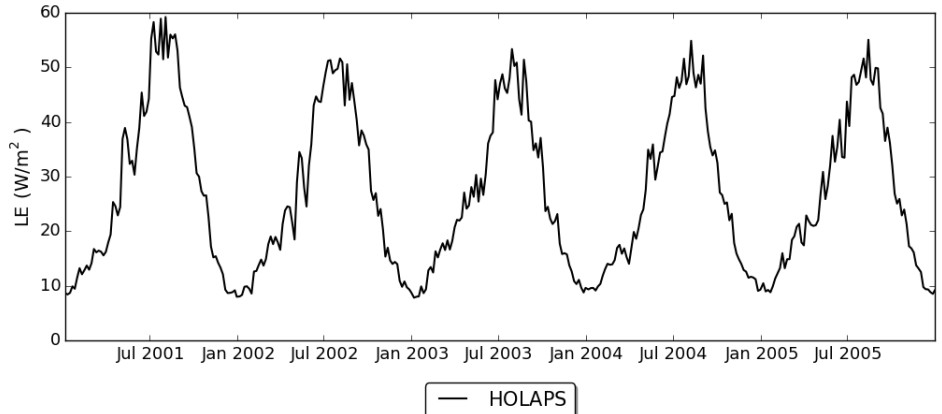


Figure 8: Temporal variability of the area averaged LE for 5-day HOLAPS over the TP.



Table 2. Correlation coefficient (R) between averaged LE and NDVI, precipitation over the TP
for the time 2001-2005.

|  | LandFlux-EVAL | HOLAPS | PM$_{SRB-PU}$ | PT$_{SRB-PU}$ | SEBS$_{SRB-PU}$ | SEBS$_{Chen}$ |
|---|---|---|---|---|---|---|
| R (NDVI) | 0.89 | 0.93 | 0.81 | 0.81 | 0.7 | 0.76 |
| R (precipitation) | 0.98 | 0.96 | 0.96 | 0.96 | 0.87 | 0.94 |


## *4.2 Comparison of LE datasets against LandFlux-EVAL benchmark*

## *product*

Figure 9 presents the monthly mean scatter plots of LE between the LandFlux-EVAL
benchmark product and other products over the whole TP. The detailed statistics are
listed in Table 3. It can be seen that the model performance varies among different LE
products with statistical indices values ranging from 0.91 to 0.99 for correlation

coefficient (R), and from 2.69 to 17.02 W/m$^2$ for RMSD. Overall, the HOLAPS
appears to yield the closest agreement with the LandFlux-EVAL benchmark product,
with R higher than 0.99 and RMSD of 2.69 W/m$^2$. In addition, the impacts of NDVI,
precipitation and elevation on the estimate of LE are also investigated. Figure 10
shows the comparison results over different NDVI thresholds. Table 4 lists the
corresponding statistics including R and RMSD. The performance of HOLAPS is
stable over different NDVI intervals, with RMSD less than 5.1 W/m$^2$. PT$_{SRB-PU}$ and
PM$_{SRB-PU}$ perform similarly with highest RMSD appearing at the lowest NDVI interval
[0, 0.15], and the RMSD of PT$_{SRB-PU}$ decreases with the increase of NDVI. Both
SEBS$_{SRB-PU}$ and SEBS$_{Chen}$ seem to overestimate LE over all NDVI intervals, with
RMSD ranging from 11.09 W/m$^2$ to 24.94 W/m$^2$. The comparison results over
different precipitation thresholds are shown in Figure 11 and Table 5. Similar to the
response to NDVI, the HOLAPS also has stable performances over different
precipitation intervals, with RMSD less than 4.91W/m$^2$. PT$_{SRB-PU}$ and PM$_{SRB-PU}$
slightly overestimate LE over the areas with low precipitation values [0, 2 mm], while
SEBS$_{SRB-PU}$ and SEBS$_{Chen}$ overestimate LE among all precipitation intervals. Figure 12
and Table 6 present the comparison results over the areas with different elevations. In
general, the elevation has no strong impacts on the HOLAPS, which has R value
higher than 0.97 and RMSD lower than 5.56 W/m$^2$ over all the elevation intervals.
PT$_{SRB-PU}$ and PM$_{SRB-PU}$ have similar performance, with overestimation of LE in areas
with high elevation [5000 m, 6000 m]. Relatively low R values for PT$_{SRB-PU}$ and
PM$_{SRB-PU}$ are also found over areas with low elevations [1000 m, 3000 m]. SEBS$_{SRB-PU}$
and SEBS$_{Chen}$ both overestimate LE over all elevation intervals. Overall, the HOLAPS
LE has stable performance over different NDVI, precipitation and elevation values.
PT$_{SRB-PU}$ and PM$_{SRB-PU}$ have very similar performance. The SEBS$_{SRB-PU}$ has the highest
uncertainty over areas with low NDVI and precipitation and high elevation, while the
highest uncertainty for SEBS$_{Chen}$ occurs in areas with high NDVI and precipitation and
low elevation.

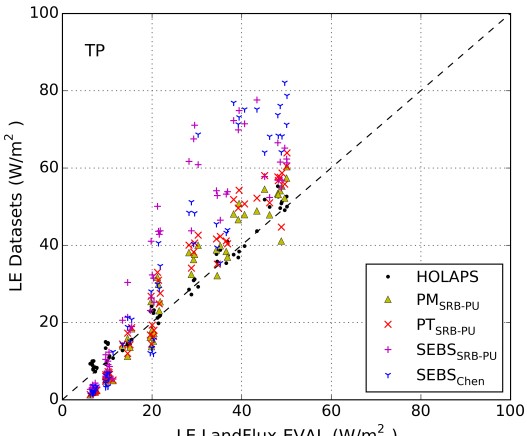

Figure 9: The monthly mean scatter plots of LE between the LandFlux-EVAL benchmark
product and other products over the whole TP.

Table 3. Statistics of the LE comparisons between the LandFlux-EVAL benchmark product
and other products over the whole TP.

| | HOLAPS | | PM$_{SRB-PU}$ | | PT$_{SRB-PU}$ | | SEBS$_{SRB-PU}$ | | SEBS$_{Chen}$ | |
|---|---|---|---|---|---|---|---|---|---|---|
| | R | RMSD (W/m$^2$) | R | RMSD (W/m$^2$) | R | RMSD (W/m$^2$) | R | RMSD (W/m$^2$) | R | RMSD (W/m$^2$) |
| Tibetan Plateau | 0.99 | 2.69 | 0.98 | 5.68 | 0.98 | 7.12 | 0.91 | 17.02 | 0.96 | 16.36 |


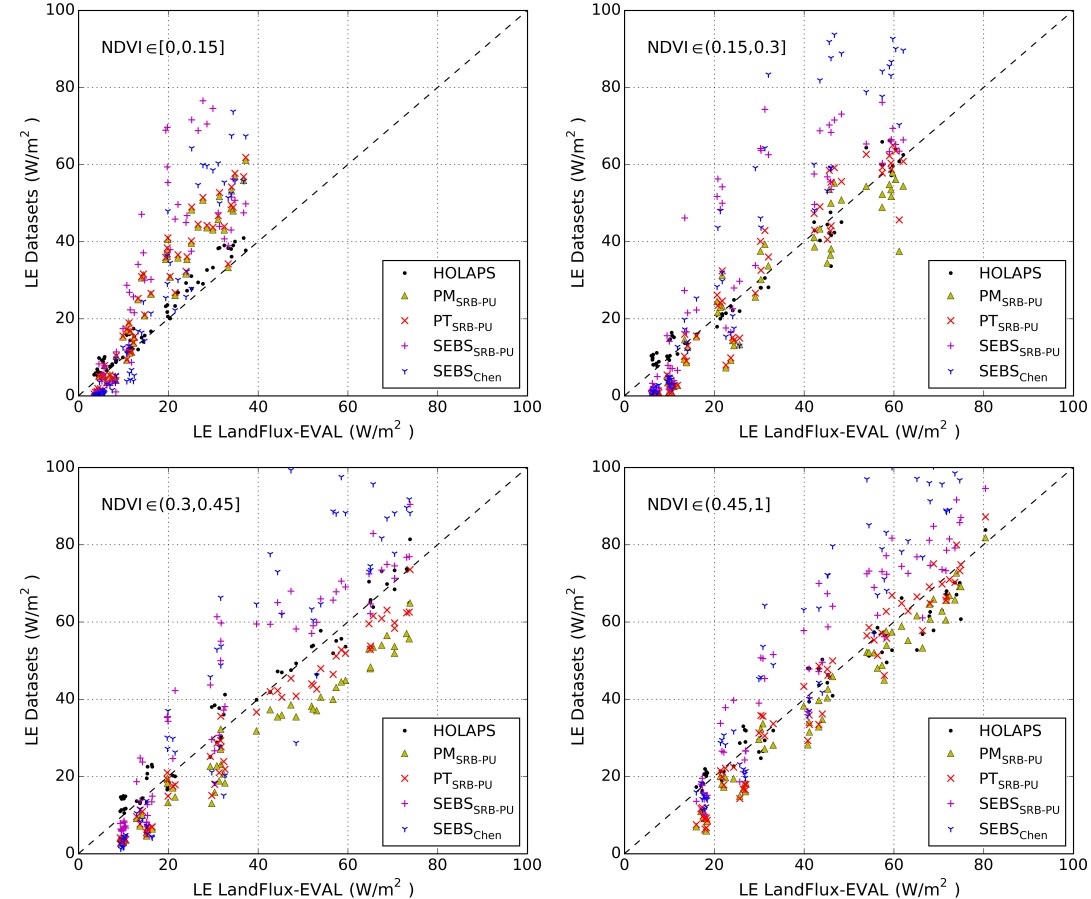


Figure 10: The monthly mean scatter plots of LE between the LandFlux-EVAL benchmark
product and other products over different NDVI thresholds.

Table 4. Statistics of the LE comparisons between the LandFlux-EVAL benchmark product
and other products over different NDVI thresholds.

| | HOLAPS | | PM$_{SRB-PU}$ | | PT$_{SRB-PU}$ | | SEBS$_{SRB-PU}$ | | SEBS$_{Chen}$ | |
|---|---|---|---|---|---|---|---|---|---|---|
| | R | RMSD (W/m$^2$) | R | RMSD (W/m$^2$) | R | RMSD (W/m$^2$) | R | RMSD (W/m$^2$) | R | RMSD (W/m$^2$) |

| | | | | | | | | | |
|---|---|---|---|---|---|---|---|---|---|
| NDVI ∈ [0, 0.15] | 0.99 | 3.27 | 0.96 | 11.42 | 0.96 | 11.89 | 0.84 | 20.93 | 0.95 | 15.38 |
| NDVI ∈ (0.15, 0.3] | 0.98 | 3.59 | 0.95 | 7.46 | 0.96 | 7.09 | 0.88 | 16.42 | 0.94 | 20.09 |
| NDVI ∈ (0.3, 0.45] | 0.98 | 4.08 | 0.99 | 10.88 | 0.99 | 7.4 | 0.94 | 11.43 | 0.92 | 17.6 |
| NDVI ∈ (0.45,1] | 0.97 | 5.1 | 0.98 | 7.1 | 0.98 | 6.21 | 0.95 | 11.87 | 0.95 | 19.11 |

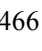




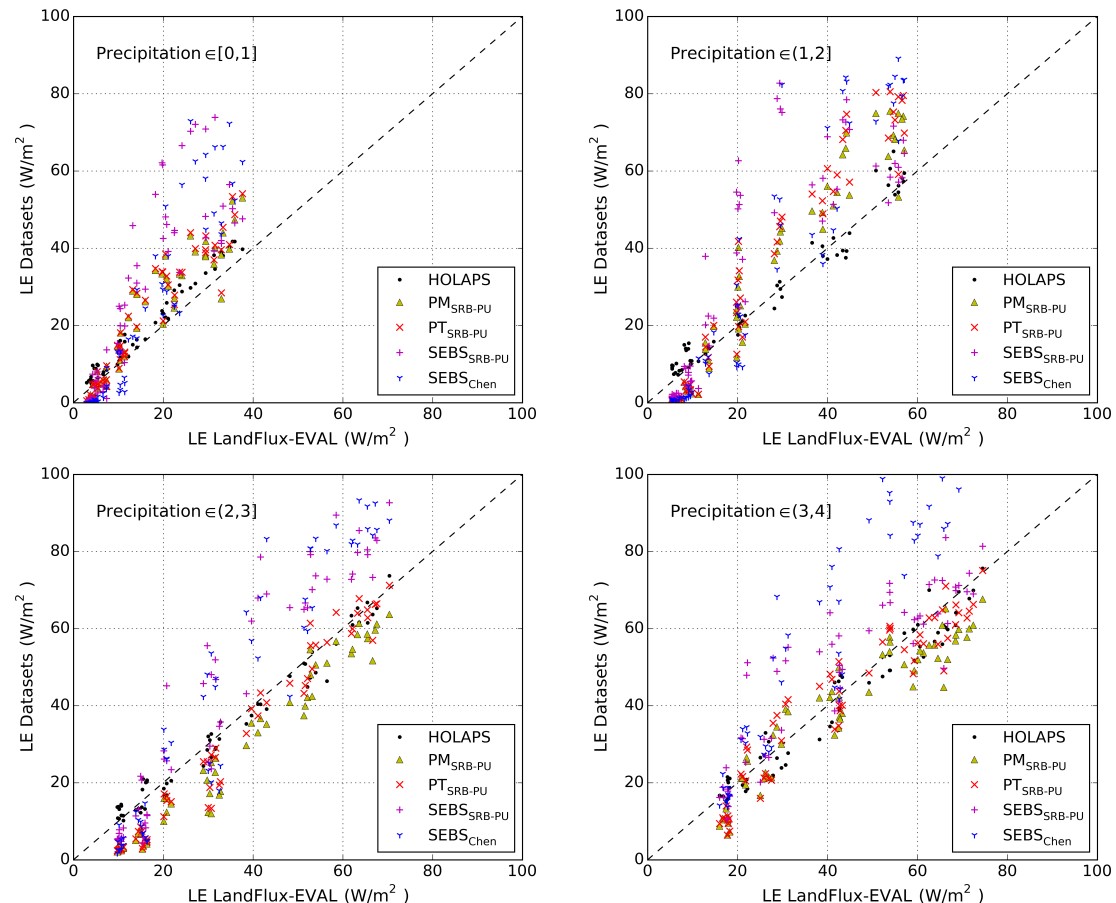



Figure 11: The monthly mean scatter plots of LE between the LandFlux-EVAL benchmark
product and other products over different precipitation thresholds.


Table 5. Statistics of the LE comparisons between the LandFlux-EVAL benchmark product
and other products over different precipitation thresholds.

| | HOLAPS | | $PM_{SRB-PU}$ | | $PT_{SRB-PU}$ | | $SEBS_{SRB-PU}$ | | $SEBS_{Chen}$ | |
|---|---|---|---|---|---|---|---|---|---|---|
| | R | RMSD (W/m$^2$) | R | RMSD (W/m$^2$) | R | RMSD (W/m$^2$) | R | RMSD (W/m$^2$) | R | RMSD (W/m$^2$) |
| Precipitation ∈ [0, 1] | 0.99 | 3.97 | 0.95 | 8.08 | 0.95 | 8.56 | 0.86 | 19.5 | 0.94 | 15.96 |
| Precipitation ∈ (1, 2] | 0.98 | 3.48 | 0.97 | 11.05 | 0.98 | 13.52 | 0.83 | 20 | 0.95 | 17.9 |

| | | | | | | | | | |
|---|---|---|---|---|---|---|---|---|---|
| Precipitation ∈ (2, 3] | 0.99 | 3.36 | 0.98 | 9.21 | 0.98 | 7.45 | 0.96 | 14.89 | 0.96 | 16.26 |
| Precipitation ∈ (3, 4] | 0.97 | 4.91 | 0.95 | 7.82 | 0.95 | 6.68 | 0.89 | 11.09 | 0.94 | 24.94 |

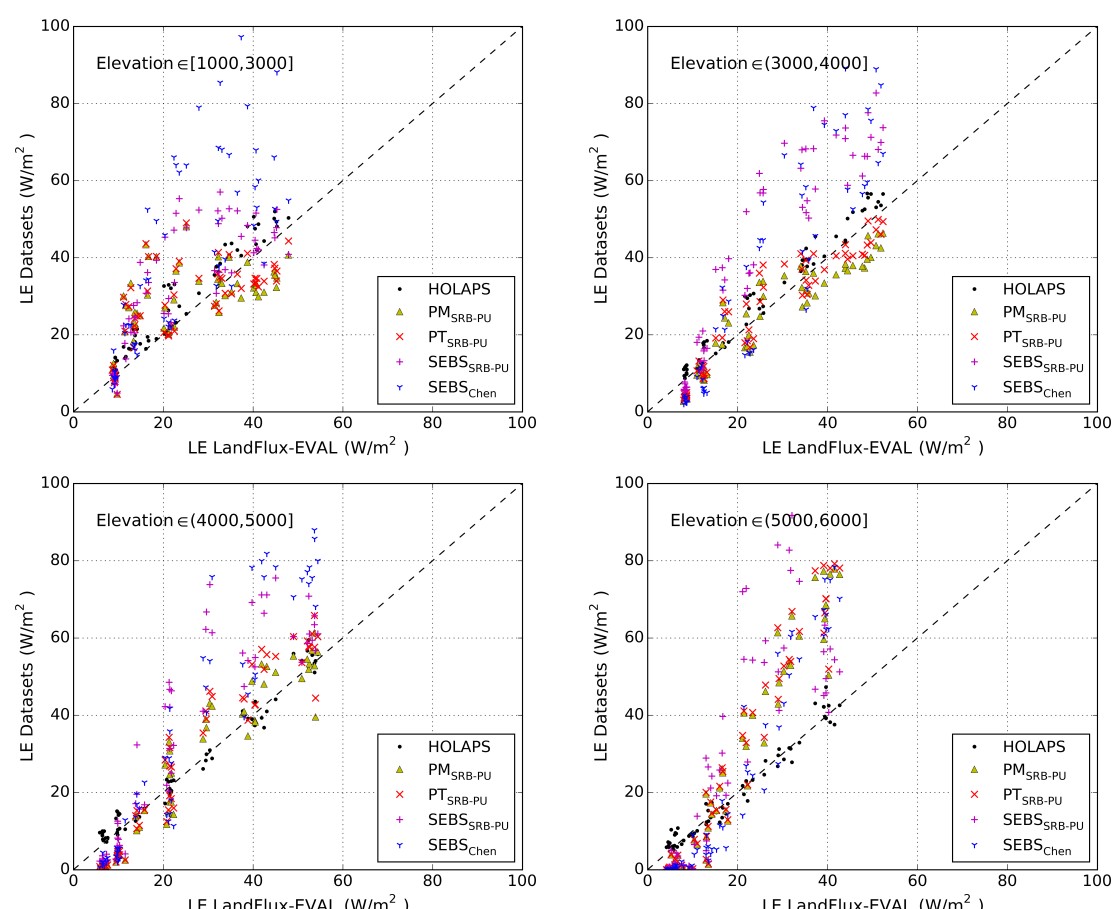


Figure 12: The monthly mean scatter plots of LE between the LandFlux-EVAL benchmark
product and other products over different elevation thresholds.


Table 6. Statistics of the LE comparisons between the LandFlux-EVAL benchmark product
and other products over different elevation thresholds.

| | HOLAPS | | PM$_{SRB-PU}$ | | PT$_{SRB-PU}$ | | SEBS$_{SRB-PU}$ | | SEBS$_{Chen}$ | |
|---|---|---|---|---|---|---|---|---|---|---|
| | R | RMSD (W/m$^2$) | R | RMSD (W/m$^2$) | R | RMSD (W/m$^2$) | R | RMSD (W/m$^2$) | R | RMSD (W/m$^2$) |
| Elevation ∈ [1000, 3000] | 0.97 | 5.56 | 0.64 | 10.24 | 0.69 | 9.94 | 0.79 | 12.92 | 0.76 | 22.53 |
| Elevation ∈ (3000, 4000] | 0.99 | 4.06 | 0.94 | 5.71 | 0.95 | 5.05 | 0.93 | 20.02 | 0.91 | 17.77 |
| Elevation ∈ (4000, 5000] | 0.99 | 2.72 | 0.96 | 6.4 | 0.97 | 7.56 | 0.9 | 15.93 | 0.95 | 17.76 |
| Elevation ∈ (5000, 6000] | 0.98 | 2.45 | 0.97 | 16.82 | 0.97 | 17.6 | 0.83 | 21.39 | 0.96 | 15.24 |


*4.3 Discussion on the different performance of the LE datasets over TP*

The spatial and temporal inter-comparisons of different global LE datasets over the TP suggest that there are large differences among different datasets. The LandFlux-EVAL benchmark product was found to agree well with observation-based precipitation, *in situ* measurements-validated radiation (Shi and Liang, 2013a) and *in situ* measurements-validated LE product (Shi and Liang, 2014). From this point of view, it can be served as the reference dataset. The HOLAPS is found to agree temporally and spatially well with LandFlux-EVAL benchmark product. The $PT_{SRB-PU}$ and $PM_{SRB-PU}$ have similar performance and are within the uncertainty range provided by LandFlux-EVAL product. Despite relying on the same forcing dataset, $SEBS_{SRB-PU}$ performs differently from $PT_{SRB-PU}$ and $PM_{SRB-PU}$, which is driven by the differences in the models. Since all these datasets rely on the same radiation forcing, the overestimation is due to the high sensitivity to the parameterization of resistances. Therefore, examination of the differences between the models especially the calculated resistances still needs to be conducted in the future work. In addition, for the same model, different forcing data lead to different results ($SEBS_{SRB-PU}$ and $SEBS_{Chen}$). The overestimation in both SEBS datasets suggests the high sensitivity of LE to the calculated resistances. And the different spatial patterns and magnitude between the two SEBS datasets are likely due to the different forcing datasets. These results suggest that model and forcing are equally critical for the estimation of ET. Future studies should be focused on the development of high quality forcing dataset, and the exploration of the sensitivity of each model to its forcing. This type of research could be facilitated by the HOLAPS framework. Because the components in HOLAPS are coupled through well-defined interfaces, which allows the integration of different models for estimation of ET while building on the general HOLAPS infrastructure for providing the consistent forcing data. Overall, the results presented here suggest that the validation and inter-comparison are essential before applying the global LE datasets for regional applications, especially for the areas with sparse in-situ measurements such as TP. The high spatial and temporal resolution HOLAPS demonstrator dataset provides a potential LE product for hydrological applications over TP. However, the current HOLAPS demonstrator dataset does not consider the ET over snow-covered areas. The parameterization scheme of ET over snow-covered areas will be added in HOLAPS framework to generate the next version of HOLAPS dataset.

# 5 Conclusions

This study provides a first comprehensive inter-comparison of existing LE products over the TP for the period 2001-2005. The results of the study can be summarized as follows:

1. The existing global LE products show substantial differences in spatial and temporal patterns over the TP, although all these products have been found to agree

well with FLUXNET measurements in different climate conditions.

2. The LandFlux-EVAL benchmark product as well as the HOLAPS LE show very similar spatial patterns, both with LE increasing from northwest to southeast. The other LE products ($SEBS_{SRB-PU}$, $SEBS_{Chen}$, $PT_{SRB-PU}$ and $PM_{SRB-PU}$) display different spatial patterns compared to LandFlux-EVAL LE. The differences between $SEBS_{SRB-PU}$, $SEBS_{Chen}$ and $PT_{SRB-PU}$, and the discrepancies between $SEBS_{SRB-PU}$ and $SEBS_{Chen}$ indicate the equal importance of model structure and forcing data. Nevertheless, all products capture well the seasonal variability with maximum LE in the summer and minimum LE in the winter. The HOLAPS LE was found to agree best with LandFlux-EVAL LE.

3. Further comparison against LandFlux-EVAL benchmark dataset over the whole TP and sub-regions that are decided by different intervals of NDVI, precipitation and elevation reveals that climate and surface conditions have impacts on the performances of $SEBS_{SRB-PU}$, $SEBS_{Chen}$, $PT_{SRB-PU}$ and $PM_{SRB-PU}$, which implying that the systematic deviations between these datasets are partly due to the impacts of different climate and surface conditions. Note that the HOLAPS LE product is insensitive to different climate and surface conditions over the TP, compared to other LE datasets.

Overall, there are still large uncertainties in the current global LE dataset over the TP. In order to accurately estimate LE over the TP, model calibration ad development of high accuracy forcing dataset are still needed. There is therefore a strong need for appropriate in situ flux measurements as well as other hydrological data like e.g. runoff measurements.

## Acknowledgements

This study uses the LandFlux-EVAL merged benchmark synthesis products of ETH Zurich produced under the aegis of the GEWEX and ILEAPS projects. The $SEBS_{SRB-PU}$, $PT_{SRB-PU}$ and $PM_{SRB-PU}$ LE products were obtained from the Princeton University Terrestrial Hydrology Research Group. The MODIS NDVI and snow cover, GPCP precipitation, and GMTED2010 elevation data are obtained from the Integrated Climate Data Center (ICDC, http://icdc.zmaw.de). This research was supported by the Cluster of Excellence CliSAP (EXC177), University of Hamburg, funded through the German Science Foundation (DFG), and the MPG-CAS postdoc fellowship. The authors would like to thank Stefan Hagemann for reviewing the first version of the manuscript. The authors are also very grateful to the editor and six anonymous reviewers for their valuable comments that helped improving the manuscript.

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
