# Peer review of "Comparison of satellite based evapotranspiration estimates over the Tibetan Plateau"

_Hydrology and Earth System Sciences, 2015_

## Referee Comment (RC1) · Anonymous Referee #1 · 14 Feb 2016

The authors present a short comparison study that compares evapotranspiration (ET) products over the topographically complex Tibetan Plateau. While accurate ET products on regions of TP are a challenge and thus demand attention, I am left a bit confused about the purpose of this paper and the value it presents to the scientific community. I feel that merely plotting a comparison of the data products without much further discussion does not warrant a publication on its own. The manuscript is well written, and the topic is very suitable for HESS. I think that major revisions could greatly improve the paper and make it a good contribution to this important topic.

General Comments

The paper is very short and there is a limited amount of in-depth analysis that is being done to compare 5 different data-sets/ approaches to a reference synthesis product.
[Figure]

The main conclusion seems to be that different algorithms that use different input data produce different results and that HOLAPS is somewhat better in comparison to the reference data set of which we don't know, how good it is on the Tibetan Plateau.

One major problem I see in the analysis is that the datasets and reference products have very different spatial and temporal resolutions, which makes a meaningful comparison problematic. This is exacerbated by the fact that ET on the semi-arid to arid TP is strongly dependent on influences of the Asian summer monsoon, which propagates northward during the summer and reaches different areas at different times (or not at all on northern TP), where dry westerlies dominate. I would suggest that the analysis should be included.

I think it would be beneficial if the authors could add a discussion that 1) tries to assess why the products produce the results that are presented here. 2) discuss the reference product and its validity on TP or at least point to such a discussion if already published by other authors.

The reference dataset is monthly only, which obviously does not allow for assessment of the products below the seasonal scale. I would expect that a lot of uncertainty in ET products on TP is on diurnal and short timescales due to monsoon dynamics and complex topography.

Due to the short nature of the paper, which almost seems to be more of a technical note, I think that such a discussion could easily be warranted without making the paper too long.

Please find below specific comments, where I think some of these general challenges could be addresses.

Specific comments:

L 49: HOLAPS (Loew et al.): At this point, the cited reference is still in review and reviewers seem to be criticizing gaps in the paper. I feel that a bit more introduction to

HOLAPS, which the authors of this paper are associated with, seems needed.

L73: "However, accurate estimation of ET over TP is still a challenge due to the limited in situ observations" - This is correct, however there is a network of flux and energy balance stations on the Tibetan Plateau, of which one of the co-authors certainly has access to the data. As HOLAPS resolutions are high, I think some comparison to this data is needed.

L118: "Since are no reliable in-situ measurements available over the TP for the 118 current study period" - Ma et al 2008 (who is a coauthor) introduce a network of flux stations that was established in the 2000s. So there should be at least some estimates of LE. If these are not available for 2003, why was this period chosen? Ma et al (2009) also estimates ET for the Plateau from satellite, which could potentially be used to compare?

Section 2: It is not entirely clear to me, whether the authors use the datasets from Vinkullu and Chen or whether they calculate these themselves. Maybe this could be added in a sentence.

L 142: "over the whole TP and four sub-regions (see Figure 1)." - Why were these subregions chose? It would feel more natural to divide these subregions to reflect climate/ monsoon influence rather that just lat/ lon.

L 150-152: "The differences between SEBSSRB-PU, PTSRB-PU and PMSRB-PU are attributed to the differences of the models. But also for the same model, different forcing data lead to different results (SEBSSRB-PU and SEBSChen). These results suggest that model and forcing are equally critical for the estimation of ET (Vinukollu et al., 2011)." - This is a trivial result, but the why is important. Maybe this would be the area to add some discussion.

Figure 2 and L153: "Overall, the HOLAPS dataset is found to have good agreement with the benchmark product (LandFlux-EVAL) with similar spatial pattern of LE" - I find

this difficult to discern based on the resolution difference. I suggest to add a figure, in which HOLAPS and SEBS_Chen are spatially averaged to the same resolution to allow for a better comparison.

Figure 4: Based on the figure, I would say that there is little difference between the products with exception of SEBS, which does not seem to work well on TP. Do the authors have a comment on why that is.

L 164-166: " In general, all products capture well the seasonal variability with minimum LE in the summer and maximum LE in the winter. However, the mean values of different LE products differ substantially" - I think that as stated above summer and winter may not be the most meaningful category as ET is mainly driven by water availability from the monsoon. So it would make sense to have at least winter (cold), dry and wet (monsoon) for discussion.

Figure 5: HOLAPS seems to do considerably worse in region 4, which is the region in which the there is most moisture available, monsoon influence is strongest and which probably has the densest sensor network. While the other 3 regions are much drier and more "remote." I feel that this potential bias for wet areas/ wet season should be explored. Region 4 is the smallest region though, so that bias may be hidden in the overall comparison (due to area averaging effects).

Conclusion: The conclusions reflect the paper, but as stated above, I feel that the results at this stage should be supplemented with a more in depth discussion of processes.

Technical comments:

L18: "Land-atmosphere interactions are largely influenced by surface latent heat fluxes ..." - In my opinion LE is an important part of land-atmosphere interactions and does not influence Land atmosphere interactions.

L19 "... due to its unique and special geographical position and physical environment"

[Figure]

- This sentence does not convey any meaning, if not followed up with specifics.

L29: " with ET decreases " - with decreasing ET

L93: While PT and PM are standard, I feel SEBS warrants a citation.

---

## Referee Comment (RC2) · Anonymous Referee #2 · 16 Feb 2016

This very brief paper is aiming to perform a comparison of six existing global evapotranspiration (ET) products over the Tibetan Plateau (TP). Even though it is an important topic and within the scope of HESS, I cannot recommend this manuscript for publication and would recommend its withdrawal and resubmission after a thorough reworking. The details for this recommendation are presented below.

**Tibetan Plateau**

Even though the TP is prominently mentioned in the title and abstract, there is no TP specific discussion present in the paper except for its description in the introduction. The TP is chosen as an area for comparison of the six models, but it might just as well have been any other region. There is no discussion of TP specific features such as varied topography, ice/snow cover, vegetation characteristics, generally dry conditions,

low atmospheric pressure, etc. Similarly, there is no discussion of suitability (or not) of any of the ET products in such conditions based on the underlying assumptions of their models and the input datasets.

For example how valid are the 1 degree datasets in such heterogeneous terrain? Do they accurately reflect the changes in surface or air temperatures with changing elevation? Or, how do the different models handle snow/ice cover, which in a region like TP this might have very significant impact on the accuracy of the modelled ET? Is it treated as in as in Vinukollu et al., 2011? In reply to reviewer the authors of the Miralles et al. (2015) publication have stated that PM-MOD and PT-JPL do not have any special modification for treating snow covered areas and use the same parameterisation as for the underlying land cover. The Loew et al. (2015) manuscript also does not described how HOLAPS deals with snow cover and indeed the reviewers of that manuscript have asked for this information. However, the current manuscript does not even use the word snow once. The above two points are just examples and there are other TP specific issues which should be discussed.

Additionally, the four sub-regions of TP used for more detailed analysis appear to have been chosen arbitrarily. If there are indeed some specific reasons of why TP was split into those sub-regions, then this should be made clear and the characteristics of each sub-region should be described. Otherwise this split serves no useful purpose and no additional information is gained compared to evaluating the models over the whole TP. What's different about region 4 compared to other regions that the results are different? It would have been much more interesting to split the area based on land cover or climatic zone or any other important property.

**Six ET products**

Almost no description of the ET products is presented. It is OK to refer the reader to the original publications for the details, but at least a basic description of underlying principles of each model should be presented, together with the major differences be-

tween them. The same goes for the different meteorological and radiation forcings. For example, Vinukolly et al. (2011) on page 4007 states that SRB albedo was unrealistically low in snow conditions. How would this affect the outputs in TP? Was albedo varied seasonally in this study or kept constant? What about other inputs such a leaf area index or land cover?

There is also no clear justification of why those ET products were chosen for this study or the applicability of LandFlux-EVAL to be used as the "benchmark" ET. What is its internal variability and applicability to TP?

Additionally, the HOLAPS model is clearly favoured by the authors (who are also the authors of the manuscript describing HOLAPS) and it is presented already in the introduction as the best model. That could be fine if the manuscript is reframed as "evaluation of HOLAPS over TP". Otherwise it appears that the conclusions were reached before the study was conducted and that is not very scientific. Finally, the HOLAPS study has not been published yet in the final form so it might be a bit too early to submit this manuscript.

**Comparison**

There is severe lack of details and analysis of the results of the comparison. For example, why only spatial patterns of the ET averaged over the whole study period are shown. In an area such as PT, where presumably there are large annual variations in phenology and snow cover, a seasonal/monthly comparisons should also be presented. Also, to better understand the spatial patterns the maps of land cover, rainfall, snow cover, etc. should also be shown.

Other questions should also be explored. For example, what drives the differences between the different models. Are the differences larger in specific land covers, specific altitudes, specific time periods, etc? How could the model assumptions impact on the differences? What about other fluxes? Are the differences due to net radiation or partitioning into H/LE? This last question could be addressed by running the HOLAPS

model with the same atmospheric and radiation forcings as other models. Or are the differences due to the spatial resolution of the input datasets. In such heterogeneous terrain the errors in the 1 degree forcings must be significant. Could HOLAPS be run with the inputs resampled to 1 degree?

Finally, there are TP focused studies presented in the introduction but there is no mention in the discussion how the magnitude and distribution of ET from the 6 products compares to the ET from those studies.

**Specific comments**

L47-54 There are other models (e.g. ALEXI) which can achieve a similar spatial and temporal resolution as HOLAPS. This was mentioned by two reviewers in the open discussion of the Loew et al. (2015) manuscript.

L81 The statement "especially HOAPS" should be justified or removed. This is the introduction and so far no results have been presented so this statement outlines the pre-conceived conclusion before the study is performed.

L101 What is the reference for the Surface Radiation Budget.

Section 3.1 There should be more discussion about the causes of the spatial patterns and the expected spatial patterns. Also it seems that there is a large seasonal difference in ET. Therefore spatial patterns split into seasons/months should be also shown and discussed.

L143 It appears that the sub-regions were chosen arbitrarily. Or do they have any differentiating characteristics? If not then it would be more informative if they were chosen based of a split in LC/climatic zones/altitude, etc. Each sub-region should be described.

L150 Patterns produced by PT and PM models are very similar, while SEBS outputs different fluxes. This should be elaborated on.
[Figure]

L151-152 What are the differences (apart from spatial resolution) between the forcings. Which one is more suitable for TP.

L159-161 PT and PM models are also within the range.

L164-165 Minimum ET is in winter, maximum is in summer.

L170-174 This seems to be important but is not elaborated on at all. Could this mean that SEBS estimates are potentially more accurate while LandFLux-EVAL is not really accurate enough to be used for benchmarking? LandFlux-EVAL also reports that its ET is towards the lower boundary of other studies (p3713 Mueller et al, 2013).

Figure 2 HOLAPS and SEBS-Chen maps should be resampled to 1 degree to allow simpler visual comparison.

L202-206 What is different about region 4 compared to other regions and the whole TP.

L207-208 This sentence does not bring anything useful to the discussion. The quantification and reduction of uncertainties might be a subject of future studies but at least those uncertainties should be described and discussed.

L209-211 There is no discussion in the results regarding the spatial resolution of any of the products or their applicability in TP environment, so this statement is unsubstantiated.
* * *

---

## Referee Comment (RC3) · Anonymous Referee #3 · 23 Feb 2016

**Comments from one reviewer**

In this work, the authors explored six available ET products based on different approaches to provide a detailed cross comparison over the Tibetan Plateau. The results are interesting, which all products capture well the seasonal variability. Moreover, regarding the spatial pattern, the High Resolution Land Surface Parameters form Space (HOLAPS) ET demonstrator dataset agrees best with LandFlux-EVAL dataset (a benchmark ET product from the Global Energy and Water Cycle Experiment). It is useful to use the HOLAPS dataset to understand the land-atmosphere-biosphere interaction over the Tibetan Plateau. Although the manuscript is written fluently, the quality of the English language and grammar needs further improvement. Thus, I recommend the manuscript to be accepted by the journal after some minor revisions. Specific comments: "Figure 1: Map of the location and topography of the Tibetan Plateau, and
four sub-regions used in this study." Question: Why and how divide TP into four subregions? Please provide scientific evidence. Methods: The manuscript would benefit of further descriptions of the related methods. Discussion: The discussion seems missing and the results with an attempt of discussion. The discussion need to be fully revised and expanded.

Line 34ïijŽ"and-atmosphere-biosphere". SuggestionïijŽland-atmosphere-biosphere. Lines 46-48ïijŽ"However, the spatial and temporal ...". SuggestionïijŽPlease check this sentence, especially "monthly means".

Lines 50-51ïijŽ"resOlution"; "surface". SuggestionïijŽResolution; Surface.

Line 53ïijŽIs HOLAPS currently the only approach or datasets?

Line 63ïijŽ"highest and largest plateau in the world ...." SuggestionïijŽTP is the highest but not largest (area) plateau in the world. The largest plateau is Brazilian Plateau, which contains five million Km2. However, TP has three million Km2. Please revise it.

Line 74ïijŽ"situ" should be in italics "situ"

Line 90ïijŽPlease write the full name of WACMOS-ET When the first appeared in the manuscript.

Lines 99-100ïijŽ"These datasets are SEBSSRB-PU, PTSRB-PU and PMSRB-PU, which are respectively based on PM, PT, and SEBS algorithms ...." SuggestionïijŽ"These datasets are SEBSSRB-PU, PTSRB-PU and PMSRB-PU, which are respectively based on SEBS, PT and PM algorithms ...."

Line 116ïijŽ"be found in (Loew et al., 2015). " should be changed as "be found in the reference of Loew et al. (2015). "

Lines 164-165ïijŽ" all products capture well the seasonal variability with minimum LE in the summer and maximum LE in the winter. "SuggestionïijŽAccording to Figure 4, all products capture well the seasonal variability with minimum LE in the winter and

HESSD
maximum LE in the summer. Also in abstract (lines 25-26).

Lines 204ïijŽ"ïijŽthe benchmark dataset (Figure 4)" SuggestionïijŽthe benchmark dataset (Figure 5)

---

## Referee Comment (RC4) · Anonymous Referee #4 · 26 Feb 2016

General comments: The evapotranspiration plays an important role in land-atmosphere interaction especially for the Tibetan Plateau which has a unique geophysical location and complex terrain. Based on six ET products produced by different models and forcing data, a cross comparison was made in this paper. It is found that HOLAPS dataset was very similar to the LandFlux-EVAL dataset and thus has the potential for the application over the TP area. The whole paper was concise and well organized. Basically, following rigorous thoughts and correct methods, the results are reasonable. The paper could be improved after taking following comments into account.

Specific comments: 1.The paper found the superiority of HOLAPS dataset if take LandFlux-EVAL dataset as a reference. It's too general to ascribe this superiority to different models or input forcing data. Although some references were listed in the paper, it's necessary to make a detailed discussion about the different physical processes

in each ET model. In other words, the author should not only give the results but also explain the reason.

2.P4, L118-120, I can not agree with this statement. It's no problem to make crosscomparison with LandFlux-EVAL dataset. Indeed, there are some in-situ measurements from 2001 to 2005 over the Tibetan Plateau.

3.P5, L153-157, It's difficult to say that similar spatial patterns exists between HOLAPS and LandFlux-EVAL. Neither can I see the LE corresponds well with the elevation from Fig. 2.

4.P5, L157-159, Please explain this statement much more clearly. What do you want to tell from Fig. 3?

5. What's your criterion to divide the whole Tibetan Plateau into four sub-regions? Does this kind of division make any sense? Why SEBS performs better in region 4 than other three regions?

---

## Referee Comment (RC5) · Anonymous Referee #5 · 1 Mar 2016

The manuscript presented a comparison study of satellite based evapotranspiration (ET) estimates over the Tibetan Plateau (TP). The estimation of ET over TP is important in many aspects and there are no accurate ET products available over TP for scientific applications yet. The current study provides a detailed analysis of six potential ET products and highlights a newly developed ET product (HOLAPS). The study concludes that the land-atmosphere interaction studies over the TP would benefit from the high resolution HOLAPS dataset. In general, the manuscript is well written, concise, and is valuable to the scientific community. It has the potential to be accepted for publication after several questions below are answered.

1. The description of HOLAPS is limited. Different from PM, PT, and SEBS, the HOLAPS is a newly developed ET product. Instead of 'refer to a reference paper', a more detailed description of HOLAPS is needed.

[Figure]

2. Why is the HOLAPS only available for the years between 2001 and 2005? For practical scientific applications, such limited time periods are not enough.

3. In addition, validation against in-situ measurements is still needed before application of the satellite-based ET product. As far as I know, there are some in-situ measurements available over the TP. Why not validate these products especially HOLAPS with the in-situ measurements? It would add significant value to the manuscript.

4. Page1, Line 25-26: the description is different from the results that are shown in figure 4. The ET should be minimum in the winter while maximum in the summer.

---

## Referee Comment (RC6) · Anonymous Referee #6 · 6 Mar 2016

**1   General comments**

The paper focuses on the comparison of different models and input data for estimating evapotranspiration (ET) in the Tibetan Plateau. A specific aim is to evaluate the HOLAPS data set, which is estimated based on remote sensing retrievals. The paper is well structured and properly written. The introduction is quite short and does not include a satisfying literature review. The discussion of the results lacks the addressing of reasons for differences in the data sets. A more in depth analysis would increase the information content of the paper tremendously. Instead the authors make the very general statment that differences in the ET products arise from differences in model assumptions and input data. I like the idea to provide a temporal and spatially highly resolved ET data set, which is based on remote sensing retrievals. I consider

the manuscript as valuable and relevant for HESS, but I recommend to return the paper with major revisions to the authors.

**2 Specific comments**

Introduction:

- In my opinion different evapotranspiration estimates based on satellite products should be discussed in the introduction - for that I think a more extensive literature review is needed. Strengths and weaknesses of different approaches should be mentioned, especially with reference to the HOLAPS data set, which is very extensively discussed in the introduction. E.g., MODIS-ET (Mu, Qiaozhen et al. 2007, 2011) (8 d temporal and 1 km spatial resolution); among others.

- L56 - L60: Here you mention very detailed results from another study which not necessarily fit into the introduction, rather it belongs to the discussion part. Furthermore, the Loew paper is still under review.

Data and methods:

- L95: briefly introduce the different assumptions and parameterizations

- What is the reasoning for dividing the study domain into the 4 regions chosen - wouldn't a devision based on e.g., morphological (e.g., elevation) or climatological (e.g., precipitation) similar regions make more sense?

Results and discussion:

- Whereas the temporal resolution of the HOLAPS data was aggregated to the temporal resolution of the LandFlux-EVAL data set the spatial resolution was not. This makes the comparison of both data sets very difficult.

- What do the white areas in Figure 2 - HOLAPS depict? Couldn't ET be estimated within this regions, if yes why?

- L161: I don't understand what selt-consistency means in that manner. Please elaborate a little bit on that.

- An analysis with regard to flux measurements would be very helpful. Chen, a co-author of this study, seems to have access to such data. A comparison of the ET products with this flux observations in the Tibetan Plateau should included in this paper already, not like stated in the conclusions in further studies.

- Please discuss the contradiction that SEBS compares well to flux observations (L171), whereas it doesn't fit to LandFlux-EVAL (L173). Potential reasons: scale mismatch between flux observations and ET products, missing processes in satellite retrievals.

- L208: Could the mentioned uncertainty which is partly stemming from the input data be further assessed by comparing/analyzing the input data from different sources (PU and Chen)?

- The different assumptions behind the evapotranspiration estimates should be discussed to get an idea where the differences between the products stem from.

- An analysis of sub-monthly ET would be interesting. Most of the data sets are on daily resolution. Did you have a closer look at this time scale?

[Figure]

**3   Technical corrections**

- L25: capture the seasonal variability well

- L34: the land-atmosphere-biosphere

- L81-82: The globally existing ET products like HOLAPS have a great potential for hydrological studiesover the TP.

- L157: The Figure → Figure

- L202: The relatively poorer ... - I do not understand this sentence.

- Figure 4: The fontsize in the legend is very small and thus hard to read. Please explain what is shown in the upper and lower graph.

- L230: maximum LE in winter and minimum LE in summer

- Figure 5: The symbols for HOLAPS and PM are hard to distinguish.

---

## Author Comment (AC1) · 26 Apr 2016

The responses for all referees' comments are provided in the attached supplement in a single document.

Please also note the supplement to this comment:
http://www.hydrol-earth-syst-sci-discuss.net/hess-2015-551/hess-2015-551-AC1-supplement.pdf

---

## Author Comment (AC4) · 26 Apr 2016

**Response to Referee #1's Comments**

*The authors present a short comparison study that compares evapotranspiration (ET) products over the topographically complex Tibetan Plateau. While accurate ET products on regions of TP are a challenge and thus demand attention, I am left a bit confused about the purpose of this paper and the value it presents to the scientific community. I feel that merely plotting a comparison of the data products without much further discussion does not warrant a publication on its own. The manuscript is well written, and the topic is very suitable for HESS. I think that major revisions could greatly improve the paper and make it a good contribution to this important topic.*

Response: We thank you very much for the constructive comments and suggestions. The aim of the current study is to figure out how different (spatially and temporally) the existing commonly used ET products are over TP. These global ET products have been validated against FLUXNET tower measurements and widely used for various applications. However, how different of these products over regional scale especially over TP are still unknown. To address this question, we carried out this study and would also like to highlight the newly developed HOLAPS product. We fully agree with your comments that further discussions on the differences of these ET products are needed. We have added new analyses and discussion on the revised manuscript according to your comments below. In the following, we provide an item-by-item response to your specific comments. Your comments are written in italic black color; our responses are shown in upright font blue color.

*General Comments*

*The paper is very short and there is a limited amount of in-depth analysis that is being done to compare 5 different data-sets/ approaches to a reference synthesis product.*

Response: Thank you for the comments. We have added new analyses and discussion on the differences between these products.

*The main conclusion seems to be that different algorithms that use different input data produce different results and that HOLAPS is somewhat better in comparison to the reference data set of which we don't know, how good it is on the Tibetan Plateau.*

Response: Yes, you are right. Actually, we cannot conclude that which ET product performs best over the TP, since there is no in-situ scale validation in this study due to the availability of in-situ measurements in the current study. Although LandFlux-EVAL has not been validated against in-situ measurements over the TP, it is the most comprehensive synthesis ET product over the TP and it provides the 25th and 75th percentile of multi-datasets. In addition, the aim of the study is not to evaluate existing ET products, but to compare and discuss the differences of these products.

*One major problem I see in the analysis is that the datasets and reference products have very different spatial and temporal resolutions, which makes a meaningful comparison problematic.*

*This is exacerbated by the fact that ET on the semi-arid to arid TP is strongly dependent on influences of the Asian summer monsoon, which propagates northward during the summer and reaches different areas at different times (or not at all on northern TP), where dry westerlies dominate. I would suggest that the analysis should be included.*

Response: We did not resample all the products to same spatial resolution, because we thought such aggregations would also introduce uncertainties, which would influence the final comparison as well. Since other reviewers also posed the same question, we have resampled these products into same grid size, and quantify the differences between them in the revised manuscript. The suggestion of investigating the influence of Asian monsoon is really a good idea. We have conducted the analysis and added the discussion into the revised manuscript.

*I think it would be beneficial if the authors could add a discussion that 1) tries to assess why the products produce the results that are presented here. 2) discuss the reference product and its validity on TP or at least point to such a discussion if already published by other authors.*

Response: Thanks for the suggestions. We have added more discussion on the differences and introduce more about the reference dataset. As we replied before, although LandFlux-EVAL has not been validated against in-situ measurements over the TP, it is the most comprehensive synthesis ET product over the TP. It includes model outputs, reanalysis datasets and satellite-based products. In addition, it provides the 25th and 75th percentile of multi-datasets, which shows the uncertainty range of the existing ET datasets. We have added these descriptions in the revised manuscript.

*The reference dataset is monthly only, which obviously does not allow for assessment of the products below the seasonal scale. I would expect that a lot of uncertainty in ET products on TP is on diurnal and short timescales due to monsoon dynamics and complex topography.*

Response: Yes, you are right. That is actually the advantage of introducing HOLAPS, which provides ET at high spatial and temporal resolution. We have investigated the HOLAPS performances at daily scale in the revised study. However, we cannot conclude which product is the best one, due to the missing of in-situ validation. The discussion about the differences and possible reasons are added in the revised manuscript.

*Due to the short nature of the paper, which almost seems to be more of a technical note, I think that such a discussion could easily be warranted without making the paper too long.*

Response: According to your constructive comments, we have made further analyses and discussions in the revised manuscript.

*Please find below specific comments, where I think some of these general challenges could be addresses.*

*Specific comments:*

*L 49: HOLAPS (Loew et al.): At this point, the cited reference is still in review and reviewers seem to be criticizing gaps in the paper. I feel that a bit more introduction to HOLAPS, which the authors of this paper are associated with, seems needed.*

Response: Yes, The HOLAPS paper (Loew et al., 2016) is still under review. We are currently working on the revision of that paper, and believe that the revised HOLAPS paper will be highly improved based on the comments and suggestions from the reviewers. To make the readers easily follow the paper, we have added more information on HOLAPS in the revised paper.

*L73: "However, accurate estimation of ET over TP is still a challenge due to the limited in situ observations" - This is correct, however there is a network of flux and energy balance stations on the Tibetan Plateau, of which one of the co-authors certainly has access to the data. As HOLAPS resolutions are high, I think some comparison to this data is needed.*

Response: To be honest, we have tried our best, however, the validation of the different ET products against flux tower measurements is not possible at the current stage due to: a) the access to suitable in-situ measurements is not possible; b) spatial representativeness of the existing towers for areas of only several square kilometers. Therefore, we decide to conduct a comprehensive validation study in the future when HOLAPS dataset is extended to longer period and increased to higher spatial resolution.

*L118: "Since are no reliable in-situ measurements available over the TP for the current study period" - Ma et al 2008 (who is a coauthor) introduce a network of flux stations that was established in the 2000s. So there should be at least some estimates of LE. If these are not available for 2003, why was this period chosen? Ma et al (2009) also estimates ET for the Plateau from satellite, which could potentially be used to compare?*

Response: Thanks for the comments. The time 2001-2005 is chosen because HOLAPS demonstrator dataset is current only available during this period due to the availability of radiation dataset. However, we cannot get access to the suitable in-situ measurements for conducting the validation. In addition, due to the temporal and spatial resolution of other ET products, it is not appropriate to compare them against in-situ measurements. Therefore, we will validate HOLAPS in the future once HOLAPS dataset is extended to longer period and increased to higher spatial resolution.

*Section 2: It is not entirely clear to me, whether the authors use the datasets from Vinkullu and Chen or whether they calculate these themselves. Maybe this could be added in a sentence.*

Response: Thanks for pointing this out. We used existing products from Vinkulu and Chen, rather than calculating them by ourselves. We have clarified this in the revised manuscript.

*L 142: "over the whole TP and four sub-regions (see Figure 1)." - Why were these subregions chose? It would feel more natural to divide these subregions to reflect climate/ monsoon influence rather that just lat/ lon.*

Response: Since other reviewers also have the same question on the dividing of TP into four regions, we have re-divided the TP region based on different thresholds of NDVI, precipitation

and elevation. The corresponding results have been analyzed and added into the revised manuscript.

*L 150-152: "The differences between SEBSSRB-PU, PTSRB-PU and PMSRB-PU are attributed to the differences of the models. But also for the same model, different forcing data lead to different results (SEBSSRB-PU and SEBSChen). These results suggest that model and forcing are equally critical for the estimation of ET (Vinukollu et al., 2011)." - This is a trivial result, but the why is important. Maybe this would be the area to add some discussion.*

Response: Thanks for the suggestion, we have added more discussion on the differences of these ET products in the revised manuscript.

*Figure 2 and L153: "Overall, the HOLAPS dataset is found to have good agreement with the benchmark product (LandFlux-EVAL) with similar spatial pattern of LE" - I find this difficult to discern based on the resolution difference. I suggest to add a figure, in which HOLAPS and SEBS_Chen are spatially averaged to the same resolution to allow for a better comparison.*

Response: Thanks, the HOLAPS and SEBS$_{chen}$ have been resampled to the same grid size as the other products, and the results are shown in the revised manuscript.

*Figure 4: Based on the figure, I would say that there is little difference between the products with exception of SEBS, which does not seem to work well on TP. Do the authors have a comment on why that is.*

Response: The different performance between SEBS and the other products have been discussed and explained in the revised manuscript.

*L 164-166: " In general, all products capture well the seasonal variability with minimum LE in the summer and maximum LE in the winter. However, the mean values of different LE products differ substantially" - I think that as stated above summer and winter may not be the most meaningful category as ET is mainly driven by water availability from the monsoon. So it would make sense to have at least winter (cold), dry and wet (monsoon) for discussion.*

Response: Thanks for the comments. In the revised manuscript, we also compared the ET products over different seasons.

*Figure 5: HOLAPS seems to do considerably worse in region 4, which is the region in which the there is most moisture available, monsoon influence is strongest and which probably has the densest sensor network. While the other 3 regions are much drier and more "remote." I feel that this potential bias for wet areas/ wet season should be explored. Region 4 is the smallest region though, so that bias may be hidden in the overall comparison (due to area averaging effects).*

Response: We have re-divide the TP based on NDVI, precipitation and elevation. The new results have been analyzed and added to replace the old four sub-region results.

*Conclusion: The conclusions reflect the paper, but as stated above, I feel that the results at this stage should be supplemented with a more in depth discussion of processes.*

Response: As stated above, further analyses and more discussion have been added in the revised manuscript.

*Technical comments:*

*L18: "Land-atmosphere interactions are largely influenced by surface latent heat fluxes ..." - In my opinion LE is an important part of land-atmosphere interactions and does not influence Land atmosphere interactions.*

Response: The sentence has been changed to: 'The knowledge of latent heat flux can help to better describe the complex mechanisms and interactions between land and atmosphere.'

*L19 "... due to its unique and special geographical position and physical environment" - This sentence does not convey any meaning, if not followed up with specifics. L29: " with ET decreases " - with decreasing ET*

Response: The sentence 'due to….physical environment' has been removed from the revised manuscript. 'with ET decreases' has been changed to 'with decreasing ET'.

*L93: While PT and PM are standard, I feel SEBS warrants a citation.*

Response: The citation has been added in the revised manuscript.

**Response to Referee #2's Comments**

*This very brief paper is aiming to perform a comparison of six existing global evapotranspiration (ET) products over the Tibetan Plateau (TP). Even though it is an important topic and within the scope of HESS, I cannot recommend this manuscript for publication and would recommend its withdrawal and resubmission after a thorough reworking. The details for this recommendation are presented below.*

Response: We thank you very much for the constructive and helpful comments. We believe that the revised manuscript has been highly improved according to your comments and suggestions. In the following, we provide an item-by-item response to your specific comments. Your comments are written in italic black color; our responses are shown in upright font blue color.

*Tibetan Plateau*

*Even though the TP is prominently mentioned in the title and abstract, there is no TP specific discussion present in the paper except for its description in the introduction. The TP is chosen as an area for comparison of the six models, but it might just as well have been any other region. There is no discussion of TP specific features such as varied topography, ice/snow cover, vegetation characteristics, generally dry conditions, low atmospheric pressure, etc. Similarly, there is no discussion of suitability (or not) of any of the ET products in such conditions based on the underlying assumptions of their models and the input datasets.*

Response: Thanks a lot for the comments. The detailed description of the TP as you suggested have been added in the manuscript. More detailed introduction (suitability) and analysis of the ET products, as well as the discussion on the differences of these products are added in the revised manuscript.

*For example how valid are the 1 degree datasets in such heterogeneous terrain? Do they accurately reflect the changes in surface or air temperatures with changing elevation? Or, how do the different models handle snow/ice cover, which in a region like TP this might have very significant impact on the accuracy of the modelled ET? Is it treated as in as in Vinukollu et al., 2011? In reply to reviewer the authors of the Miralles et al. (2015) publication have stated that PM-MOD and PT-JPL do not have any special modification for treating snow covered areas and use the same parameterisation as for the underlying land cover. The Loew et al. (2015) manuscript also does not described how HOLAPS deals with snow cover and indeed the reviewers of that manuscript have asked for this information. However, the current manuscript does not even use the word snow once. The above two points are just examples and there are other TP specific issues which should be discussed.*

Response: We do not intend to evaluate these ET products and find out the most accurate product, because LandFlux-EVAL is not the truth data. Instead, we would like to compare and discuss the differences of these products. In addition, we would like to highlight the advantages of HOLAPS, which can provide ET at high spatial and temporal resolution. Through the comparison, we tried to figure out how to estimate ET over TP more accurately, like improving

or even developing new ET scheme, develop new forcing dataset. We have made a deep discussion on the differences of these products. We also analyzed the influence of NDVI, precipitation and elevation on the performance of these products. The 'with or without' snow schemes in these products are also discussed in the revised manuscript.

*Additionally, the four sub-regions of TP used for more detailed analysis appear to have been chosen arbitrarily. If there are indeed some specific reasons of why TP was split into those sub-regions, then this should be made clear and the characteristics of each sub-region should be described. Otherwise this split serves no useful purpose and no additional information is gained compared to evaluating the models over the whole TP. What's different about region 4 compared to other regions that the results are different? It would have been much more interesting to split the area based on land cover or climatic zone or any other important property.*

Response: Since other reviewers also have the same question on the dividing of TP into four regions, we have re-divided the TP region based on different thresholds of NDVI, precipitation and elevation. The corresponding results have been analyzed and added into the revised manuscript.

*Six ET products*

*Almost no description of the ET products is presented. It is OK to refer the reader to the original publications for the details, but at least a basic description of underlying principles of each model should be presented, together with the major differences between them. The same goes for the different meteorological and radiation forcings. For example, Vinukolly et al. (2011) on page 4007 states that SRB albedo was unrealistically low in snow conditions. How would this affect the outputs in TP? Was albedo varied seasonally in this study or kept constant? What about other inputs such a leaf area index or land cover?*

Response: Thanks for the comments and questions. The detailed description and analysis of the feasibility of these schemes over TP have been added. Besides, the discussions about the forcing datasets such as radiation are also added in the revised manuscript. In the current study, we cannot answer how the albedo influences ET. Theoretically, if the albedo was underestimated, the ET would be overestimated. In HOLAPS, the albedo is from MODIS, and it varies with seasons. The LAI is also seasonally variable, but land cover is kept constant. For further details, please refer to Loew et al., (2016).

*There is also no clear justification of why those ET products were chosen for this study or the applicability of LandFlux-EVAL to be used as the "benchmark" ET. What is its internal variability and applicability to TP?*

Response: These global ET products have been validated against FLUXNET tower measurements and widely used for various applications. However, how different and accurate of these products over regional scale especially over TP are still unknown. To address this question, we carried out this study and would like to highlight the newly developed HOLAPS product. Although LandFlux-EVAL has not been validated against in-situ measurements over the TP, it is the most comprehensive synthesis ET product over the TP and it provides the 25th and 75th

percentile of multi-datasets. In addition, the aim of the study is not to evaluate existing ET products, but to compare and discuss the differences of these products. The justification of the use of these products has been added in the revised manuscript.

*Additionally, the HOLAPS model is clearly favoured by the authors (who are also the authors of the manuscript describing HOLAPS) and it is presented already in the introduction as the best model. That could be fine if the manuscript is reframed as "evaluation of HOLAPS over TP". Otherwise it appears that the conclusions were reached before the study was conducted and that is not very scientific. Finally, the HOLAPS study has not been published yet in the final form so it might be a bit too early to submit this manuscript.*

Response: Thanks a lot for the comment. The current study is rather a comparison than an evaluation paper. We have removed the improper description of HOLAPS in the introduction. Yes, The HOLAPS paper (Loew et al., 2016) is still under review. We are currently working on the revision of that paper, and believe that the revised HOLAPS paper will be highly improved based on the comments and suggestions from the reviewers. To make the readers easily follow the paper, we have added more information on HOLAPS in the revised paper.

*Comparison*

*There is severe lack of details and analysis of the results of the comparison. For example, why only spatial patterns of the ET averaged over the whole study period are shown. In an area such as PT, where presumably there are large annual variations in phenology and snow cover, a seasonal/monthly comparisons should also be presented. Also, to better understand the spatial patterns the maps of land cover, rainfall, snow cover, etc. should also be shown.*

Response: Thank you for the suggestions. The seasonal comparisons have been added, and we also showed the time series of different products for the new defined regions.

*Other questions should also be explored. For example, what drives the differences between the different models. Are the differences larger in specific land covers, specific altitudes, specific time periods, etc? How could the model assumptions impact on the differences? What about other fluxes? Are the differences due to net radiation or partitioning into H/LE? This last question could be addressed by running the HOLAPS model with the same atmospheric and radiation forcings as other models. Or are the differences due to the spatial resolution of the input datasets. In such heterogeneous terrain the errors in the 1 degree forcings must be significant. Could HOLAPS be run with the inputs resampled to 1 degree?*

Response: Thank you very much for the constructive questions. We have explored the influence of NDVI, elevation, and rainfall on the comparison of different products. The different model structure and forcing datasets are also discussed in the revised manuscript. We are using the exiting products generated by Vinkulu and Chen, which were already based on different forcings. If we run HOLAPS with the same forcing as others, then we have to rerun all other models with same forcing as well, which is beyond the scope of the current work, and WACMOS-ET and GEWEX LandFlux projects (Michel et al., 2016; McCabe et al., 2016) have already done the similar work over global scale. HOLAPS can be run with inputs resampled to 1

degree. However, instead of rerunning HOLAPS, we resampled the HOLAPS ET product to 1 degree and compared it to other products.

*Finally, there are TP focused studies presented in the introduction but there is no mention in the discussion how the magnitude and distribution of ET from the 6 products compares to the ET from those studies.*

Response: Thanks for the comment. The discussion of the difference between the 6 products and other existing ET products are discussed in the revised manuscript.

*Specific comments*

*L47-54 There are other models (e.g. ALEXI) which can achieve a similar spatial and temporal resolution as HOLAPS. This was mentioned by two reviewers in the open discussion of the Loew et al. (2015) manuscript.*

Response: The ALEXI model has been introduced in the revised manuscript.

*L81 The statement "especially HOAPS" should be justified or removed. This is the introduction and so far no results have been presented so this statement outlines the pre-conceived conclusion before the study is performed.*

Response: Thanks, done.

*L101 What is the reference for the Surface Radiation Budget.*

Response: The reference has been added.

*Section 3.1 There should be more discussion about the causes of the spatial patterns and the expected spatial patterns. Also it seems that there is a large seasonal difference in ET. Therefore spatial patterns split into seasons/months should be also shown and discussed.*

Response: The spatial patterns have been analyzed through comparison to NDVI, rainfall, elevation and snow map. In addition, the seasonal/monthly comparison has also been added into the revised manuscript.

*L143 It appears that the sub-regions were chosen arbitrarily. Or do they have any differentiating characteristics? If not then it would be more informative if they were chosen based of a split in LC/climatic zones/altitude, etc. Each sub-region should be described.*

Response: As stated above, the TP has been re-divided based on NDVI, precipitation and elevation.

*L150 Patterns produced by PT and PM models are very similar, while SEBS outputs different fluxes. This should be elaborated on.*

Response: The reasons of high SEBS output have been discussed in the revised manuscript.

*L151-152 What are the differences (apart from spatial resolution) between the forcings. Which one is more suitable for TP.*

Response: The discussion about the forcings has been added in the revised manuscript.

*L159-161 PT and PM models are also within the range.*

Response: The description has been modified.

*L164-165 Minimum ET is in winter, maximum is in summer.*

Response: Changed, thanks.

*L170-174 This seems to be important but is not elaborated on at all. Could this mean that SEBS estimates are potentially more accurate while LandFLux-EVAL is not really accurate enough to be used for benchmarking? LandFlux-EVAL also reports that its ET is towards the lower boundary of other studies (p3713 Mueller et al, 2013).*

Response: That is a good point. However, it is hard to conclude which product is the best one due to the missing of validation against in-situ measurements. Even though we have the in-situ measurements, the sparse in-situ stations cannot guarantee the best performance of the product over non in-situ measurements areas. As the TP condition is very complex, probably none of the existing ET models work well over TP. Therefore, the aim is to quantify the differences of the existing products, and outlook the future steps on how to better estimate ET over TP. In addition, even though LandFlux-EVAL has not been validated against in-situ measurements over the TP, it is the most comprehensive synthesis ET product over the TP and it provides the 25th and 75th percentile of multi-datasets, implying the uncertainty range of the existing ET products. On the point of this view, it is reasonable to take LandFlux-EVAL as a reference dataset.

*Figure 2 HOLAPS and SEBS-Chen maps should be resampled to 1 degree to allow simpler visual comparison.*

Response: Done. The comparison results are shown in the revised manuscript.

*L202-206 What is different about region 4 compared to other regions and the whole TP.*

Response:  The TP has been re-divided and analyzed based on different NDVI, precipitation and elevation.

*L207-208 This sentence does not bring anything useful to the discussion. The quantification and reduction of uncertainties might be a subject of future studies but at least those uncertainties should be described and discussed.*

Response: The quantification and discussion on the differences of these products have been added in the revised manuscript.

*L209-211 There is no discussion in the results regarding the spatial resolution of any of the products or their applicability in TP environment, so this statement is unsubstantiated.*

Response: The statement has been removed.

References:

Michel, D., Jiménez, C., Miralles, D. G., Jung, M., Hirschi, M., Ershadi, A., Martens, B., McCabe, M. F., Fisher, J. B., Mu, Q., Seneviratne, S. I., Wood, E. F., and Fernández-Prieto, D.: The WACMOS-ET project – Part 1: Tower-scale evaluation of four remote-sensing-based evapotranspiration algorithms, Hydrology and Earth System Sciences, 20, 803-822, 10.5194/hessd-12-10739-2015, 2016.

McCabe, M. F., Ershadi, A., Jimenez, C., Miralles, D. G., Michel, D., and Wood, E. F.: The GEWEX LandFlux project: evaluation of model evaporation using tower-based and globally gridded forcing data, Geoscientific Model Development, 9, 283-305, 10.5194/gmd-9-283-2016, 2016.

**Response to Referee #3's Comments**

*In this work, the authors explored six available ET products based on different approaches to provide a detailed cross comparison over the Tibetan Plateau. The results are interesting, which all products capture well the seasonal variability. Moreover, regarding the spatial pattern, the High Resolution Land Surface Parameters form Space (HOLAPS) ET demonstrator dataset agrees best with LandFlux-EVAL dataset (a benchmark ET product from the Global Energy and Water Cycle Experiment). It is useful to use the HOLAPS dataset to understand the land-atmosphere-biosphere interaction over the Tibetan Plateau. Although the manuscript is written fluently, the quality of the English language and grammar needs further improvement. Thus, I recommend the manuscript to be accepted by the journal after some minor revisions.*

Response: We thank you very much for the encouragement and suggestions. The English and grammar of the manuscript have been further improved by a native English speaker. In the following, we provide an item-by-item response to your specific comments. Your comments are written in italic black color; our responses are shown in upright font blue color.

*Specific comments: "Figure 1: Map of the location and topography of the Tibetan Plateau, and four sub-regions used in this study." Question: Why and how divide TP into four sub- regions? Please provide scientific evidence.*

Response: Since other reviewers also have the same question on the dividing of TP into four regions, we have re-divided the TP region based on different thresholds of NDVI, precipitation and elevation. The corresponding results have been analyzed and added into the revised manuscript.

*Methods: The manuscript would benefit of further descriptions of the related methods.*

Response: The description of the methods has been added in the revised manuscript.

*Discussion: The discussion seems missing and the results with an attempt of discussion. The discussion need to be fully revised and expanded.*

Response: Thank you for the comments. We have added new analyses and discussion on the differences between these products.

*Line 34:"and-atmosphere-biosphere". Suggestion:land-atmosphere-biosphere.*

Response: Done.

*Lines 46-48:"However, the spatial and temporal . . .". Suggestion:Please check this sentence, especially "monthly means".*

Response: 'monthly means' has been changed to 'monthly mean'.

*Lines 50-51:"resOlution"; "surface". Suggestion:Resolution; Surface.*

Response: The purpose of highlighting O in 'resOlution' is due to the HOLAPS abbreviation.

*Line 53:Is HOLAPS currently the only approach or datasets?*

Response: The sentence has been changed to 'HOLAPS is actually a framework that can provide surface energy and water fluxes at comparably high spatial and temporal resolutions'

*Line 63:"highest and largest plateau in the world . . ." Suggestion:TP is the highest but not largest (area) plateau in the world. The largest plateau is Brazilian Plateau, which contains five million Km2. However, TP has three million Km2. Please revise it.*

Response: Thanks for pointing this out. The description has been revised.

*Line 74:"situ" should be in italics "situ"*

Response: Done.

*Line 90:Please write the full name of WACMOS-ET When the first appeared in the manuscript.*

Response: Done.

*Lines 99-100:"These datasets are SEBSSRB-PU, PTSRB-PU and PMSRB-PU, which are respectively based on PM, PT, and SEBS algorithms ..." Suggestion:"These datasets are SEBSSRB-PU, PTSRB-PU and PMSRB-PU, which are respectively based on SEBS, PT and PM algorithms . . ."*

Response: Thanks, done.

*Line 116:"be found in (Loew et al., 2015). " should be changed as "be found in the reference of Loew et al. (2015). "*

Response: Done.

*Lines 164-165:" all products capture well the seasonal variability with minimum LE in the summer and maximum LE in the winter. " Suggestion:According to Figure 4, all products capture well the seasonal variability with minimum LE in the winter and maximum LE in the summer. Also in abstract (lines 25-26).*

Response: Thanks, done.

*Lines 204:":the benchmark dataset (Figure 4)" Suggestion:the benchmark dataset (Figure 5)*

Response: Done.

**Response to Referee #4's Comments**

*General comments: The evapotranspiration plays an important role in land-atmosphere interaction especially for the Tibetan Plateau which has a unique geophysical location and complex terrain. Based on six ET products produced by different models and forcing data, a cross comparison was made in this paper. It is found that HOLAPS dataset was very similar to the LandFlux-EVAL dataset and thus has the potential for the application over the TP area. The whole paper was concise and well organized. Basically, following rigorous thoughts and correct methods, the results are reasonable. The paper could be improved after taking following comments into account.*

Response: We thank you very much for the encouragement. In the following, we provide an item-by-item response to your specific comments. Your comments are written in italic black color; our responses are shown in upright font blue color.

*Specific comments:*

*1.The paper found the superiority of HOLAPS dataset if take LandFlux-EVAL dataset as a reference. It's too general to ascribe this superiority to different models or input forcing data. Although some references were listed in the paper, it's necessary to make a detailed discussion about the different physical processes in each ET model. In other words, the author should not only give the results but also explain the reason.*

Response: Thanks for the comments. The description and analyses of different ET models have been added. In addition, we have also added new analyses and discussion on the differences between these products.

*2.P4, L118-120, I can not agree with this statement. It's no problem to make cross- comparison with LandFlux-EVAL dataset. Indeed, there are some in-situ measurements from 2001 to 2005 over the Tibetan Plateau.*

Response: To be honest, we have tried our best, however, the validation of the different ET products against flux tower measurements is not possible at the current stage due to: a) the access to suitable in-situ measurements is not possible; b) spatial representativeness of the existing towers for areas of only several square kilometers. Therefore, we decide to conduct a comprehensive validation study in the future when HOLAPS dataset is extended to longer period and increased to higher spatial resolution.

*3.P5, L153-157, It's difficult to say that similar spatial patterns exists between HOLAPS and LandFlux-EVAL. Neither can I see the LE corresponds well with the elevation from Fig. 2.*

Response: We have added new analyses in the revised manuscript. The HOLAPS has been resampled to the same grid size as LandFlux-EVAL. Therefore the differences between spatial patterns can be better explored. In addition, the inter-comparison between different products based on stratification of elevation has also been added in the revised manuscript.

*4.P5, L157-159, Please explain this statement much more clearly. What do you want to tell from Fig. 3?*

Response: Since the LandFlux-EVAL is a benchmarking product, and its 25th-percentile and 75th-percentile give the uncertainties range of many datasets. Therefore, it is meaningful to explore if the magnitude of used ET product is within these ranges. The statement has been improved in the revised manuscript.

*5.What's your criterion to divide the whole Tibetan Plateau into four sub-regions? Does this kind of division make any sense? Why SEBS performs better in region 4 than other three regions?*

Response: Since other reviewers also have the same question on the dividing of TP into four regions, we have re-divided the TP region based on different thresholds of NDVI, precipitation and elevation. The corresponding results have been analyzed and added into the revised manuscript.

**Response to Referee #5's Comments**

*The manuscript presented a comparison study of satellite based evapotranspiration (ET) estimates over the Tibetan Plateau (TP). The estimation of ET over TP is important in many aspects and there are no accurate ET products available over TP for scientific applications yet. The current study provides a detailed analysis of six potential ET products and highlights a newly developed ET product (HOLAPS). The study concludes that the land-atmosphere interaction studies over the TP would benefit from the high resolution HOLAPS dataset. In general, the manuscript is well written, concise, and is valuable to the scientific community. It has the potential to be accepted for publication after several questions below are answered.*

Response: We thank you very much for the encouragement. In the following, we provide an item-by-item response to your specific comments. Your comments are written in italic black color; our responses are shown in upright font blue color.

*1. The description of HOLAPS is limited. Different from PM, PT, and SEBS, the HOLAPS is a newly developed ET product. Instead of 'refer to a reference paper', a more detailed description of HOLAPS is needed.*

Response: Thanks. The description and analyses of different ET models have been added in the revised manuscript.

*2. Why is the HOLAPS only available for the years between 2001 and 2005? For practical scientific applications, such limited time periods are not enough.*

Response: The HOLAPS itself is a framework, it has the potential to generate longterm surface and water fluxes. The current analysis is based on the HOLAPS demonstrator product, which was generated only for 2001-2005 due to the availability of solar radiation data.

*3. In addition, validation against in-situ measurements is still needed before application of the satellite-based ET product. As far as I know, there are some in-situ measurements available over the TP. Why not validate these products especially HOLAPS with the in-situ measurements? It would add significant value to the manuscript.*

Response: To be honest, we have tried our best, however, the validation of the different ET products against flux tower measurements is not possible at the current stage due to: a) the access to suitable in-situ measurements is not possible; b) spatial representativeness of the existing towers for areas of only several square kilometers. Therefore, we decide to conduct a comprehensive validation study in the future when HOLAPS dataset is extended to longer period and increased to higher spatial resolution.

*4. Page1, Line 25-26: the description is different from the results that are shown in figure 4. The ET should be minimum in the winter while maximum in the summer.*

Response: Thanks, done.

**Response to Referee #6's Comments**

*1 General comments*

*The paper focuses on the comparison of different models and input data for estimating evapotranspiration (ET) in the Tibetan Plateau. A specific aim is to evaluate the HOLAPS data set, which is estimated based on remote sensing retrievals. The paper is well structured and properly written. The introduction is quite short and does not include a satisfying literature review. The discussion of the results lacks the addressing of reasons for differences in the data sets. A more in depth analysis would increase the information content of the paper tremendously. Instead the authors make the very general statment that differences in the ET products arise from differences in model assumptions and input data. I like the idea to provide a temporal and spatially highly resolved ET data set, which is based on remote sensing retrievals. I consider the manuscript as valuable and relevant for HESS, but I recommend to return the paper with major revisions to the authors.*

Response: We thank you very much for the encouragement and suggestions. The introduction has been highly improved through the integration of recent publications. The description and analyses of different ET models have been added in the revised manuscript. In addition, we have also added new analyses and discussion on the differences between these products. In the following, we provide an item-by-item response to your specific comments. Your comments are written in italic black color; our responses are shown in upright font blue color.

*2 Specific comments*

*Introduction:*

*• In my opinion different evapotranspiration estimates based on satellite products should be discussed in the introduction - for that I think a more extensive literature review is needed. Strengths and weaknesses of different approaches should be mentioned, especially with reference to the HOLAPS data set, which is very extensively discussed in the introduction. E.g., MODIS-ET (Mu, Qiaozhen et al. 2007, 2011) (8 d temporal and 1 km spatial resolution); among others.*

Response: Thanks for the suggestions. The introduction has been highly improved through the integration of recent publications. The description and analyses of different ET models have been added in the revised manuscript. The other high temporal and spatial ET products like MODIS-ET, ALEXI have also been introduced in the introduction.

*• L56 - L60: Here you mention very detailed results from another study which not necessarily fit into the introduction, rather it belongs to the discussion part. Furthermore, the Loew paper is still under review.*

Response: We have removed the improper description of HOLAPS in the introduction. Yes, The HOLAPS paper (Loew et al., 2016) is still under review. We are currently working on the revision

of that paper, and believe that the revised HOLAPS paper will be highly improved based on the comments and suggestions from the reviewers.

*Data and methods:*

*• L95: briefly introduce the different assumptions and parameterizations*

Response: Done.

*• What is the reasoning for dividing the study domain into the 4 regions chosen - wouldn't a devision based on e.g., morphological (e.g., elevation) or climatological (e.g., precipitation) similar regions make more sense?*

Response: Since other reviewers also have the same question on the dividing of TP into four regions, we have re-divided the TP region based on different thresholds of NDVI, precipitation and elevation. The corresponding results have been analyzed and added into the revised manuscript.

*Results and discussion:*

*• Whereas the temporal resolution of the HOLAPS data was aggregated to the temporal resolution of the LandFlux-EVAL data set the spatial resolution was not. This makes the comparison of both data sets very difficult.*

Response: The HOLAPS and SEBS have been resampled to the same grid size as LandFlux-EVAL. The spatial comparison between these products and LandFlux-EVAL have also been improved in the revised manuscript.

*• What do the white areas in Figure 2 - HOLAPS depict? Couldn't ET be estimated within this regions, if yes why?*

Response: The white areas are snow and ice covered areas. In the current HOLAPS framework, the snow covered areas are not included for calculation. However, we are currently working on integrate the ET scheme over snow into the HOLAPS framework.

*• L161: I don't understand what selt-consistency means in that manner. Please elaborate a little bit on that.*

Response: The phrase 'to test their self-consistency' has been removed in the revised manuscript.

*• An analysis with regard to flux measurements would be very helpful. Chen, a co-author of this study, seems to have access to such data. A comparison of the ET products with this flux observations in the Tibetan Plateau should included in this paper already, not like stated in the conclusions in further studies.*

Response: To be honest, we have tried our best, however, the validation of the different ET products against flux tower measurements is not possible at the current stage due to: a) the access to suitable in-situ measurements is not possible; b) spatial representativeness of the existing towers for areas of only several square kilometers. Therefore, we decide to conduct a comprehensive validation study in the future when HOLAPS dataset is extended to longer period and increased to higher spatial resolution.

• *Please discuss the contradiction that SEBS compares well to flux observations (L171), whereas it doesn't fit to LandFlux-EVAL (L173). Potential reasons: scale mismatch between flux observations and ET products, missing processes in satellite retrievals.*

Response: Thanks for the suggestions. The possible reasons have been discussed in the revised manuscript.

• *L208: Could the mentioned uncertainty which is partly stemming from the input data be further assessed by comparing/analyzing the input data from different sources (PU and Chen)?*

Response: That is a good idea. However, we are using the exiting ET products generated by Vinkulu and Chen, which were based on different forcings. And we do not have these forcing datasets. In addition, we expect that the comparison of forcings would show the differences among these forcing datasets. The interesting study would be investigating the sensitivity of SEBS to these forcings. However, it is not the aim of the current study.

• *The different assumptions behind the evapotranspiration estimates should be discussed to get an idea where the differences between the products stem from.*

Response: Thanks for the suggestion. The differences of the ET products have been further discussed in the manuscript in terms of model assumptions and special conditions over TP.

• *An analysis of sub-monthly ET would be interesting. Most of the data sets are on daily resolution. Did you have a closer look at this time scale?*

Response: Yes, we agree that the sub-monthly such as daily analysis is more interesting. The daily HOLAPS has been plotted and shown in the revised manuscript.

*3 Technical corrections*

• *L25: capture the seasonal variability well*

Response: Done.

• *L34: the land-atmosphere-biosphere*

Response: Done.

• *L81-82: The globally existing ET products like HOLAPS have a great potential for hydrological studies over the TP.*

Response: Done.

*• L157: The Figure → Figure*

Response: Done.

*• L202: The relatively poorer ... - I do not understand this sentence.*

Response: In the revised manuscript, the dividing of TP has been based on NDVI, precipitation and elevation. And the new results have been analyzed and added in the revised manuscript.

*• Figure 4: The fontsize in the legend is very small and thus hard to read. Please explain what is shown in the upper and lower graph.*

Response: The fontsize has been enlarged. The description of upper and lower graph has been added in the caption.

*• L230: maximum LE in winter and minimum LE in summer*

Response: Corrected.

*• Figure 5: The symbols for HOLAPS and PM are hard to distinguish.*

Response: The symbols have been adjusted.